# Warm-up Free Policy Optimization: Improved Regret in Linear Markov Decision Processes

**Asaf Cassel**
Tel Aviv University
acassel@mail.tau.ac.il

**Aviv Rosenberg**
Google Research
avivros@google.com

## Abstract

Policy Optimization (PO) methods are among the most popular Reinforcement Learning (RL) algorithms in practice. Recently, Sherman et al. [2023a] proposed a PO-based algorithm with rate-optimal regret guarantees under the linear Markov Decision Process (MDP) model. However, their algorithm relies on a costly pure exploration warm-up phase that is hard to implement in practice. This paper eliminates this undesired warm-up phase, replacing it with a simple and efficient contraction mechanism. Our PO algorithm achieves rate-optimal regret with improved dependence on the other parameters of the problem (horizon and function approximation dimension) in two fundamental settings: adversarial losses with full-information feedback and stochastic losses with bandit feedback.

## 1 Introduction

Policy Optimization (PO) is a widely used method in Reinforcement Learning (RL) that achieved tremendous empirical success, with applications ranging from robotics and computer games [Schulman et al., 2015, 2017, Mnih et al., 2015, Haarnoja et al., 2018] to Large Language Models (LLMs; Stiennon et al. [2020], Ouyang et al. [2022]). Theoretical work on policy optimization algorithms initially considered tabular Markov Decision Processes (MDPs; Even-Dar et al. [2009], Neu et al. [2010b], Shani et al. [2020], Luo et al. [2021]), where the number of states is assumed to be finite and small. In recent years the theory was generalized to infinite state spaces under function approximation, specifically under linear function approximation in the linear MDP model [Luo et al., 2021, Dai et al., 2023, Sherman et al., 2023b,a, Liu et al., 2023].

Recently, Sherman et al. [2023a] presented the first policy optimization algorithm that achieves rate-optimal regret in linear MDPs, i.e., a regret bound of $\widetilde{O}(\text{poly}(H, d)\sqrt{K})$, where $K$ is the number of interaction episodes, $H$ is the horizon, and $d$ is the dimension of the linear function approximation. However, their algorithm requires a pure exploration warm-up phase to obtain an initial estimate of the transition dynamics. To that end, they utilize the algorithm of Wagenmaker et al. [2022b] for reward-free exploration which is not based on the policy optimization paradigm. Moreover, although this algorithm is computationally efficient, it relies on intricate estimation techniques that are hard to implement in practice and unlikely to generalize beyond linear function approximation (see discussion in section 4).

In this paper, we propose a novel contraction mechanism to avoid this costly warm-up phase. Both our contraction mechanism and the warm-up phase serve a similar purpose – ensuring that the Q-value estimates are bounded and yield "simple" policies. But, unlike the warm-up, our method is integrated directly into the PO algorithm, implemented using a simple conditional truncation of the Q-estimates, and only contributes a lower-order term to the final regret bound. Moreover, our approach is much

more efficient in practice since it does not rely on any reward-free methods, which explore the state space uniformly without taking the reward into account.

Based on this contraction mechanism, we build a new policy optimization algorithm that is simpler, more computationally efficient, easier to implement, and most importantly, improves upon the best-known regret bounds for policy optimization in linear MDPs. Our regret bound holds in two fundamental settings:

1. Adversarial losses with full-information feedback, where the loss function changes arbitrarily between episodes and is revealed to the agent entirely at the end of each episode.

2. Stochastic losses with bandit feedback, where the loss function in each episode is sampled i.i.d from some unknown fixed distribution and the agent only observes instantaneous losses in the state-action pairs that she visits.

In these settings, the best-known regret bound (by Sherman et al. [2023b]) was $\widetilde{O}(\sqrt{H^7 d^4 K})$. Our algorithm, Contracted Features Policy Optimization (CFPO), achieves $\widetilde{O}(\sqrt{H^4 d^3 K})$ regret, yielding a $\sqrt{H^3 d}$ improvement over any algorithm for the adversarial setting and matching the value iteration based approach of Jin et al. [2020b] in the stochastic setting. We conjecture that this is the best regret we can hope for without more sophisticated variance reduction techniques [Azar et al., 2017, Zanette and Brunskill, 2019, He et al., 2023, Zhang et al., 2024], that have not yet been applied to PO algorithms even in the tabular setting.[1] Ignoring logarithmic factors, the regret of CFPO leaves a gap of only $\sqrt{Hd}$ from the $\Omega(\sqrt{H^3 d^2 K})$ lower bound for linear MDPs [Zhou et al., 2021a]. Finally, our analysis relies on a novel regret decomposition that uses a notion of contracted (sub) MDP and may be of separate interest (see section 5).

## 1.1 Related work

**Policy optimization in tabular MDPs.** The regret analysis of PO methods in tabular MDPs was introduced by Even-Dar et al. [2009], which considered the case of known transitions and adversarial losses under full-information feedback. Neu et al. [2010a,b] extended their algorithms to adversarial losses under bandit feedback. Then, Shani et al. [2020] presented the first PO algorithms for the case of unknown transitions (for both stochastic and adversarial losses), and finally Luo et al. [2021] devised a PO algorithm with rate-optimal regret for the challenging case of unknown transitions with adversarial losses under bandit feedback. Since then, PO was studied in more challenging cases, e.g., delayed feedback [Lancewicki et al., 2022, 2023] and best-of-both-worlds [Dann et al., 2023].

**Other regret minimization methods in tabular MDPs.** An alternative popular method for regret minimization in tabular MDPs with adversarial losses is O-REPS [Zimin and Neu, 2013, Rosenberg and Mansour, 2019a,b, Jin et al., 2020a], which optimizes over the global state-action occupancy measures instead of locally over the policies in each state. However, this method is hard to implement in practice and does not generalize to the function approximation setting (without restrictive assumptions). For stochastic losses, optimistic methods based on Value Iteration (VI; Jaksch et al. [2010], Azar et al. [2017], Zanette and Brunskill [2019]) and Q-learning [Jin et al., 2018, Zhang et al., 2020] are known to guarantee optimal regret, which has not been established yet for adversarial losses.

**Policy optimization in linear MDPs.** While Sherman et al. [2023a] established rate-optimal regret for PO methods in linear MDPs with stochastic losses, most of the recent research focused on the case of adversarial losses with bandit feedback [Luo et al., 2021, Neu and Olkhovskaya, 2021, Dai et al., 2023, Sherman et al., 2023b, Kong et al., 2023, Liu et al., 2023, Zhong and Zhang, 2023], where rate-optimality has not been achieved yet.

**Other regret minimization methods in linear MDPs and other models for function approximation.** Unlike O-REPS methods that do not generalize to linear function approximation, value-based methods (operating under the stochastic loss assumption) are also popular in linear MDPs and have been shown to yield optimal regret [Jin et al., 2020a, Zanette et al., 2020, Wagenmaker et al., 2022a,

---

[1]Wu et al. [2022] apply variance reduction techniques to get better regret bounds in the tabular setting, but they use $L_2$-regularization instead of KL-regularization which does not align with practical PO algorithms Schulman et al. [2015, 2017].

Hu et al., 2022, He et al., 2023, Agarwal et al., 2023]. Another line of works [Ayoub et al., 2020, Modi et al., 2020, Cai et al., 2020, Zhang et al., 2021, Zhou et al., 2021a,b, He et al., 2022, Zhou and Gu, 2022] study linear mixture MDP which is a different model that is incomparable with linear MDP [Zhou et al., 2021b]. Finally, there is a rich line of works studying statistical properties of RL with more general function approximation [Munos, 2005, Jiang et al., 2017, Dong et al., 2020, Jin et al., 2021, Du et al., 2021], but these usually do not admit computationally efficient algorithms.

## 2 Problem setup

**Episodic Markov Decision Process (MDP).** A finite-horizon episodic MDP $\mathcal{M}$ is defined by a tuple $(\mathcal{X}, \mathcal{A}, x_1, \{\ell^k\}_{k=1}^K, P, H)$ with $\mathcal{X}$, a set of states, $\mathcal{A}$, a set of actions, $H$, decision horizon, $x_1 \in \mathcal{X}$, an initial state (assumed to be fixed for simplicity), $P = (P_h)_{h \in [H]}, P_h : \mathcal{X} \times \mathcal{A} \to \Delta(\mathcal{X})$, the transition probabilities, and $\{\ell^k\}_{k=1}^K$, sequence of loss functions such that $\ell^k = (\ell_h^k)_{h \in [H]}, \ell_h^k : \mathcal{X} \times \mathcal{A} \to [0, 1]$, is a horizon dependent immediate loss function for taking action $a$ at state $x$ and horizon $h$ of episode $k$. A single episode $k$ of an MDP is a sequence $(x_h^k, a_h^k, \ell_h^k(x_h^k, a_h^k))_{h \in [H]} \in (\mathcal{X} \times \mathcal{A} \times [0, 1])^H$ such that

$$\Pr[x_{h+1}^k = x' \mid x_h^k = x, a_h^k = a] = P_h(x' \mid x, a).$$

For the losses, we consider two settings: stochastic and adversarial. In the stochastic setting, there exists a fixed loss function $\ell = (\ell_h)_{h \in [H]}, \ell_h : \mathcal{X} \times \mathcal{A} \to [0, 1]$ such that $\ell^k$ is sampled i.i.d from a distribution whose expected value is defined by $\ell$, i.e., $\mathbb{E}[\ell_h^k(x, a) \mid x, a] = \ell_h(x, a)$. In the adversarial setting, the loss function sequence $\{\ell^k\}_{k=1}^K$ is chosen by an adaptive adversary.

**Linear MDP.** A linear MDP Jin et al. [2020b] satisfies all the properties of the above MDP but has the following additional structural assumptions. There is a known feature mapping $\phi : \mathcal{X} \times A \to \mathbb{R}^d$ such that $P_h(x' \mid x, a) = \phi(x, a)^\mathsf{T} \psi_h(x')$ where $\psi_h : \mathcal{X} \to \mathbb{R}^d$ are unknown parameters. Moreover, for all $h \in [H], k \in [K]$, there is an unknown vector $\theta_h^k \in \mathbb{R}^d$ such that, in the adversarial case, $\ell_h^k(x, a) = \phi(x, a)^\mathsf{T} \theta_h^k$, while in the stochastic case, $\theta_h^k = \theta_h$ and $\ell_h(x, a) = \phi(x, a)^\mathsf{T} \theta_h$. We make the following normalization assumptions, common throughout the literature:

1. $\|\phi(x, a)\| \le 1$ for all $x \in X, a \in \mathcal{A}$;
2. $\|\theta_h^k\| \le \sqrt{d}$ for all $h \in [H], k \in [K]$;
3. $\||\psi_h|(\mathcal{X})\| = \|\sum_{x \in \mathcal{X}} |\psi_h(x)|\| \le \sqrt{d}$ for all $h \in [H]$;

where $|\psi_h(x)|$ is the entry-wise absolute value of $\psi_h(x) \in \mathbb{R}^d$. We follow the standard assumption in the literature that the action space $\mathcal{A}$ is finite. In addition, for ease of mathematical exposition (e.g. Cassel et al. [2024]), we also assume that the state space $\mathcal{X}$ is finite. This allows for simple matrix notation and avoids technical measure theoretic definitions. Importantly, our results are completely independent of the state space size $|\mathcal{X}|$, both computationally and in terms of regret. Thus, there is no particular loss of generality.

**Policy and value.** A stochastic Markov policy $\pi = (\pi_h)_{h \in [H]} : [H] \times \mathcal{X} \mapsto \Delta(\mathcal{A})$ is a mapping from a step and a state to a distribution over actions. Such a policy induces a distribution over trajectories $\iota = (x_h, a_h)_{h \in [H]}$, i.e., sequences of $H$ state-action pairs. For $f : (\mathcal{X} \times \mathcal{A})^H \to \mathbb{R}$, which maps trajectories to real values, we denote the expectation with respect to $\iota$ under dynamics $P$ and policy $\pi$ as $\mathbb{E}_{P, \pi}[f(\iota)]$. Similarly, we denote the probability under this distribution by $\mathbb{P}_{P, \pi}[\cdot]$. We denote the class of stochastic Markov policies as $\Pi_M$. For any policy $\pi \in \Pi_M$, horizon $h \in [H]$ and episode $k \in [K]$ we define its loss-to-go, as

$$V_h^{k, \pi}(x) = \mathbb{E}_{P, \pi}\left[\sum_{h'=h}^H \mathbb{E}[\ell_{h'}^k(x_{h'}, a_{h'}) \mid x_{h'}, a_{h'}] \,\middle|\, x_h = x\right],$$

which is the expected loss if one starts from state $x \in \mathcal{X}$ at horizon $h$ of episode $k$ and follows policy $\pi$ onwards. Note that the inner expectation is only relevant for stochastic losses as its argument is deterministic in the adversarial setup. The performance of a policy in episode $k$, also known as its value, is measured by its expected cumulative loss $V_1^{k, \pi}(x_1)$.

**Interaction protocol and regret.** We consider a standard episodic regret minimization setting where an algorithm performs $K$ interactions with an MDP $\mathcal{M}$. For stochastic losses we consider bandit feedback, where the agent observes only the instantaneous losses along its trajectory, while for adversarial losses we consider full-information feedback, where the agent observes the full loss function $\ell^k$ in the end of episode $k \in [K]$. Concretely, at the start of each interaction/episode $k \in [K]$, the agent specifies a stochastic Markov policy $\pi^k = (\pi_h^k)_{h \in [H]}$. Subsequently, it observes the trajectory $\iota^k$ sampled from the distribution $\mathbb{P}_{P,\pi^k}$, and, either the individual episode losses $\ell_h^k(x_h^k, a_h^k), h \in [H]$ in the case of bandit feedback, or the entire loss function $\ell^k$ in the case of full-information feedback.

We measure the quality of any algorithm via its *regret* – the difference between the value of the policies $\pi^k$ generated by the algorithm and that of the best policy in hindsight, i.e.,

$$\text{Regret} = \sum_{k=1}^{K} V_1^{k,\pi^k}(x_1) - \min_{\pi \in \Pi_M} \sum_{k=1}^{K} V_1^{k,\pi}(x_1) = \sum_{k=1}^{K} V_1^{k,\pi^k}(x_1) - V_1^{k,\pi^\star}(x_1),$$

where the best policy in hindsight is denoted by $\pi^\star$ (known to be optimal even among the class of stochastic history-dependent policies).

**Notation.** Throughout the paper $\phi_h^k = \phi(x_h^k, a_h^k) \in \mathbb{R}^d$ denote the state-action features at horizon $h$ of episode $k$. In addition, $\|v\|_A = \sqrt{v^\mathsf{T} A v}$. Hyper-parameters follow the notations $\beta_z$ and $\eta_z$ for some $z$, and $\delta \in (0,1)$ denotes a confidence parameter. Finally, in the context of an algorithm, $\leftarrow$ signs refer to compute operations whereas $=$ signs define operators, which are evaluated at specific points as part of compute operations.

## 3 The role of value clipping

Before presenting our contraction technique and main results, we discuss the role that value clipping plays in regret minimization and its apparent necessity for linear MDPs. As a starting point, it is important to note that, while commonly used [Azar et al., 2017, Luo et al., 2021], value clipping is not strictly necessary in tabular MDPs. To demonstrate this, consider a fairly standard optimistic Value Iteration (VI) algorithm that constructs sample-based estimates $\hat{\ell}, \hat{P}$ with empirical error estimates $\Delta_\ell, \Delta_P$, defines exploration bonuses $b = (\Delta_\ell + H \cdot \Delta_P)$, and chooses a policy $\hat{\pi}^\star$ that is optimal in the empirical MDP whose dynamics are $\hat{P}$ and losses are $\hat{\ell} - b$. Then its single-episode regret may be decomposed as

$$V_1^{\hat{\pi}^\star}(x_1) - V_1^{\pi^\star}(x_1) = \underbrace{V_1^{\hat{\pi}^\star}(x_1) - \hat{V}_1^{\hat{\pi}^\star}(x_1)}_{(i) - \text{bias / cost of optimism}} + \underbrace{\hat{V}_1^{\hat{\pi}^\star}(x_1) - \hat{V}_1^{\pi^\star}(x_1)}_{(ii) - \text{FTL / ERM}} + \underbrace{\hat{V}_1^{\pi^\star}(x_1) - V_1^{\pi^\star}(x_1)}_{(iii) - \text{optimism}},$$

where $\hat{V}$ is the value under the empirical MDP. Now, by definition of $\hat{\pi}^\star$, we have that $(ii) \leq 0$. Now, let $\Delta \ell = \hat{\ell} - \ell, \Delta P = \hat{P} - P$. Using a standard value difference lemma (lemma 14 in appendix B) we have that $(i) \lesssim b$ and

$$(iii) = \mathbb{E}_{\hat{P},\pi^\star}\left[ \sum_{h \in [H]} \Delta \ell(x_h, a_h) - b(x_h, a_h) + \sum_{x' \in \mathcal{X}} \Delta P(x' \mid x_h, a_h) V_{h+1}^{\pi^\star}(x') \right] \tag{1}$$

$$\leq \mathbb{E}_{\hat{P},\pi^\star}\left[ \sum_{h \in [H]} \Delta_\ell(x_h, a_h) + H \Delta_P(x_h, a_h) - b(x_h, a_h) \right] = 0,$$

where the inequality also used that $V_h^{\pi^\star} \in [0, H]$. The final regret bound is concluded by summing over $k \in [K]$ and using a bound on harmonic sums. We note that a similar clipping-free method also works for tabular PO (see Cassel et al. [2024]).

Moving on to Linear MDPs, one might expect a similar approach to work. Unfortunately, the standard approach that estimates the dynamics backup operators $\psi_h, h \in [H]$ using regularized least-squares presents a significant challenge. This is because, unlike the tabular setting, the resulting estimate

$\hat{P}_h(\cdot \mid x, a) = \phi(x, a)^\mathsf{T} \widehat{\psi}_h(\cdot)$ (eq. (2)) is not guaranteed to yield a valid probability distribution, i.e., there could exist $x \in \mathcal{X}, a \in \mathcal{A}, h \in [H]$ such that

$$\|\hat{P}_h(\cdot \mid x, a)\|_1 = c > 1 \quad \text{and/or} \quad \min_{x' \in \mathcal{X}} \hat{P}_h(x' \mid x, a) < 0.$$

$\hat{P}$ is still a finite signed-measure, which is enough for the first equality in eq. (1) to hold. However, since $\mathbb{E}_{\hat{P}, \pi^\star}$ could contain negative probability terms, the inequality in eq. (1) does not hold. These negative probabilities also seem to make calculating $\hat{\pi}^\star$ computationally hard. Finally, the $\ell_1$−norm exceeding 1 may cause term $(i)$ to depend on $H$ exponentially. While some of these issues could be mitigated without clipping, we are not aware of a method that resolves all simultaneously.

The use of value clipping opens the path for an alternative value decomposition that replaces $\mathbb{E}_{\hat{P}, \pi^\star}$ in eq. (1) with $\mathbb{E}_{P, \pi^\star}$ at the cost of also replacing $V_{h+1}^{\pi^\star}$ with $\hat{V}_{h+1}^{\pi^\star}$. We thus need that $|\hat{V}_{h+1}^{\pi^\star}| \lesssim H$ for the inequality in eq. (1) to work. This is made possible using a clipping mechanism that decouples the scale of $\hat{V}_{h+1}^{\pi^\star}$ from the magnitude of the bonuses $b$, which may be much larger when the error estimates $\Delta_\ell, \Delta_P$ are large. This is typically achieved by adding $\max\{0, \cdot\}$ to the recursive formula for the value function. A similar clipping approach also works for tabular PO and VI [Azar et al., 2017, Luo et al., 2021], and even for VI in linear MDPs [Jin et al., 2020b].

However, this is not the case for PO in linear MDPs where Sherman et al. [2023a] explain that this type of value clipping leads to prohibitive complexity of the policy and value function classes, and thus sub-optimal regret. Concretely, the complexity of the soft-max policy class roughly corresponds to the number of parameters required to represent $\sum_{k \in [K]} \hat{Q}_h^k$. If $\hat{Q}_h^k(x, a) = \phi(x, a)^\mathsf{T} w_h^k$ are linear, then the sum remains linear and depends on $d$ parameters (with slightly larger magnitude). If $\hat{Q}_h^k(x, a) = \max\{0, \phi(x, a)^\mathsf{T} w_h^k\}$, the sum may, in general, have $dK$ parameters thus degrading the regret. Sherman et al. [2023a] overcome this issue using a warm-up based truncation technique. In what follows, we suggest an alternative solution that uses a novel notion of contracted features and has several advantages over their approach (see discussion at the end of section 4).

## 4 Algorithm and main result

We present Contracted Features Policy Optimization (CFPO; algorithm 1), a policy optimization routine for regret minimization in linear MDPs. The algorithm operates in epochs, each beginning when the uncertainty of the dynamics estimation shrinks by a multiplicative factor, as expressed by the determinant of the covariance matrices $\Lambda_h^k, h \in [H]$ (see line 13 for the definition of $\Lambda_h^k$ and line 4 for the epoch change condition). At the start of each epoch $e$, we reset the policy to its initial (uniform) state, and define the contracted features $\bar{\phi}_h^{k_e}, h \in [H]$ (line 6) by multiplying the original features with coefficients in the range $[0, 1]$, and thus shrinking their distance to the origin. Inspired by ideas from Zanette et al. [2020], these coefficients are chosen inversely proportional to the current uncertainty of the least squares estimators in each state-action pair, essentially degenerating the MDP in areas of high uncertainty. Inside an epoch, at episode $k$, we compute the estimated reward vector $\widehat{\theta}^k$ (line 14) and estimated dynamics backup operators $\widehat{\psi}_h^k$ (eq. (2)). Then, we use these $\widehat{\theta}^k$ and $\widehat{\psi}_h^k$ to compute our Q-value estimates with the contracted features (eq. (3)), and run an online mirror descent (OMD) update over them (eq. (5)), i.e., run a policy optimization step with respect to the contracted empirical MDP (more on this in section 5.1).

We note that the computational complexity of algorithm 1 is comparable to other algorithms for regret minimization in linear MDPs, such as LSVI-UCB [Jin et al., 2020b]. The following is our main result for algorithm 1 (see the full analysis appendix A).

**Theorem 1.** *Suppose that we run CFPO (algorithm 1) with the parameters defined in theorem 9 (in appendix A). Then, with probability at least $1 - \delta$, we have*

$$\text{Regret} = O\left(\sqrt{H^4 d^3 K \log(K) \log(KH/\delta)} + \sqrt{H^5 dK \log(K) \log|\mathcal{A}|}\right).$$

**Discussion.** Policy optimization algorithms typically entail running OMD over estimates $\hat{Q}$ of the state-action value function $Q$, as in eq. (5). The crux of the algorithm is in obtaining such estimates that satisfy an optimistic condition similar to eq. (1), while also keeping the complexity of the policy class bounded. As discussed in Sherman et al. [2023a], the latter depends on $\sum_{k' \in [k]} \hat{Q}_h^{k'}$ (eq. (3))

---
**Algorithm 1** Contracted Features PO for linear MDPs
---
1: **input**: $d, H, K, \mathcal{A}, \delta, \beta_w, \beta_b, \eta_o > 0$.
2: **initialize**: $e \leftarrow -1, \Lambda_h^1 \leftarrow I, h \in [H]$.
3: **for** episode $k = 1, 2, \ldots, K$ **do**
4:     **if** $k = 1$ **or** $\exists h \in [H], \det(\Lambda_h^k) \geq 2 \det(\Lambda_h^{k_e})$ **then**
5:         $e \leftarrow e + 1$ and $k_e \leftarrow k$.
6:         $\bar{\phi}_h^{k_e}(x, a) = \phi(x, a) \cdot \sigma\left(-\beta_w \|\phi(x, a)\|_{(\Lambda_h^{k_e})^{-1}} + \log K\right).$     $\{\sigma(z) = 1/(1 + \exp(-z))\}$
7:         $\pi_h^k(a \mid x) = 1/|\mathcal{A}|$ for all $h \in [H], a \in \mathcal{A}, x \in \mathcal{X}$.
8:     **end if**
9:     Play $\pi^k$ and observe losses $(\ell_h^k(x_h^k, a_h^k))_{h \in [H]}$ and trajectory $\iota^k = (x_h^k, a_h^k)_{h \in [H]}$.
10:     In the case of full-information feedback: observe $\theta_h^k$.
11:     Define $\hat{V}_{H+1}^k(x) = 0$ for all $x \in \mathcal{X}$.
12:     **for** $h = H, \ldots, 1$ **do**
13:         $\Lambda_h^{k+1} \leftarrow I + \sum_{\tau \in [k]} \phi_h^\tau (\phi_h^\tau)^\mathsf{T}$.
14:         $\widehat{\theta}_h^k \leftarrow \begin{cases} (\Lambda_h^k)^{-1} \sum_{\tau \in [k-1]} \phi_h^\tau \ell_h^\tau(x_h^\tau, a_h^\tau), & \text{feedback} = bandit \\ \theta_h^k, & \text{feedback} = full. \end{cases}$
15:         For any $V : \mathcal{X} \to \mathbb{R}, x \in \mathcal{X}, a \in \mathcal{A}$ define:

$$\widehat{\psi}_h^k V = (\Lambda_h^k)^{-1} \sum_{\tau \in [k-1]} \phi_h^\tau V(x_{h+1}^\tau), \tag{2}$$

$$\hat{Q}_h^k(x, a) = \bar{\phi}_h^{k_e}(x, a)^\mathsf{T}[\widehat{\theta}_h^k + \widehat{\psi}_h^k \hat{V}_{h+1}^k] - \beta_b \|\bar{\phi}_h^{k_e}(x, a)\|_{(\Lambda_h^{k_e})^{-1}}, \tag{3}$$

$$\hat{V}_h^k(x) = \sum_{a \in \mathcal{A}} \pi_h^k(a \mid x) \hat{Q}_h^k(x, a), \tag{4}$$

$$\pi_h^{k+1}(a \mid x) \propto \pi_h^k(a \mid x) \exp(-\eta_o \hat{Q}_h^k(x, a)). \tag{5}$$

16:     **end for**
17: **end for**
---

having a low dimensional representation nearly independent of $k$. Although standard unclipped estimates admit such a representation, they lack other essential properties (see discussion in section 3). On the other hand, the standard clipping method, which restricts the value to $[0, H]$ between each backup operation (see, e.g., Jin et al. [2020b]), does not admit the desired representation.

Sherman et al. [2023a] overcame this issue by employing a warm-up phase based on a reward-free pure exploration algorithm by Wagenmaker et al. [2022b] to obtain initial backup operators $\widehat{\psi}_h^0, h \in [H]$ and subsets $\bar{\mathcal{X}}_h \subseteq \mathcal{X}, h \in [H]$ such that: (i) for every $x, a \in \bar{\mathcal{X}}_h \times \mathcal{A}$ the bonuses (b in section 3), which are proportional to the estimation uncertainty of the value backup estimates, are small ($\leq 1$); and (ii) for all policies $\pi \in \Pi_M$, the probability of reaching any $x, a \notin \cup_{h \in [H]} \bar{\mathcal{X}}_h \times \mathcal{A}$ is small ($\lesssim K^{-1/2}$). To ensure that the overall value estimates remain bounded, they truncate (zero out) the Q-value estimate of these nearly unreachable state-action pairs, an operation that allows for a low-dimensional representation of the policies. Nonetheless, their warm-up approach has several drawbacks.

- It runs for $K_0 = \text{poly}(d, H)\sqrt{K}$ episodes, contributing the leading term in their regret guarantee;

- It relies on a first-order regret algorithm by Wagenmaker et al. [2022a] that is not PO-based and uses a computationally hard variance-aware Catoni estimator for robust mean estimation of the value backups, instead of the standard least-squares estimator. To maintain computational efficiency, they use an approximate version of the estimator, losing a factor of $\sqrt{d}$ in the regret;

- Still, to the best of our knowledge, even the approximate estimator must be computed using binary search methods, making it hard to apply in practical methods that typically rely on gradient-based continuous optimization techniques;

- It runs separate algorithms for each horizon $h \in [H]$, using only 1 out of $H$ samples during the warm-up phase;
- It is not reward-aware, and thus has to explore the space uniformly to ensure that the uncertainty is small for all policies, which could be highly prohibitive in practice.

Our feature contraction approach obtains the desired bounded Q-value estimates and low-complexity policy class without relying on a dedicated warm-up phase. Crucially, it only contributes a lower order term of $\mathrm{poly}(d, H) \log K$ to the regret guarantee, thus improving the overall dependence on $d$ and $H$. Additionally, it uses all samples, is easy to implement, and is reward-aware. To understand the benefit of reward-awareness, consider an MDP where at the initial state the agent has two actions, each leading to a distinct MDP. Now, suppose that both MDPs have only a single state and action for the first $H/2$ steps with one MDP incurring a loss of 1 in these steps while the other incurring 0 loss. Notice that regardless of the last $H/2$ steps, the 0 loss MDP will outperform the 1 loss MDP. Nonetheless, the reward-free warm-up, which does not observe the losses, will have to fully explore both MDPs. In contrast, our reward-aware approach would quickly stop exploring the inferior MDP, leading to better performance in practice.

## 5 Analysis

In this section, we prove the main claims of our result. For full details see appendix A. We begin by introducing the main technical tool for our contraction mechanism – the contracted MDP.

### 5.1 Contracted (sub) MDP

For any MDP $\mathcal{M} = (\mathcal{X}, \mathcal{A}, x_1, \{\ell^k\}_{k=1}^K, P, H)$ and contraction coefficients $\rho : [H] \times \mathcal{X} \times \mathcal{A} \to [0, 1]$ we define a contracted (sub) MDP $\bar{\mathcal{M}}(\rho) = (\mathcal{X}, \mathcal{A}, x_1, \{\bar{\ell}^k\}_{k=1}^K, \bar{P}, H)$ where as $\bar{\ell}_h^k(x, a) = \rho_h(x, a)\ell_h^k(x, a) \in [0, 1]$ are the contracted losses and $\bar{P}_h(x' \mid x, a) = \rho_h(x, a)P_h(x' \mid x, a) \in [0, 1]$ are the contracted (sub) probability transitions. Notice that the transitions being a sub-probability measure implies that $\sum_{x' \in \mathcal{X}} \bar{P}_h(x' \mid x, a) \leq 1$ as compared with a probability measure where this holds with equality. For any Markov policy $\pi \in \Pi_M$, let $\bar{V}_h^{k,\pi}(\cdot; \rho) : \mathcal{X} \to \mathbb{R}, h \in [H]$ be the loss-to-go (or value) functions of the contracted MDP. In particular, these may be defined by the usual backward recursion

$$\bar{V}_h^{k,\pi}(x; \rho) = \mathbb{E}_{a \sim \pi(\cdot|x)}\left[\mathbb{E}[\bar{\ell}_h^k(x, a) \mid x, a] + \sum_{x' \in \mathcal{X}} \bar{P}_h(x' \mid x, a)\bar{V}_{h+1}^{k,\pi}(x'; \rho)\right],$$

with $\bar{V}_{H+1}^{k,\pi}(x; \rho) = 0$ for all $x \in \mathcal{X}$. The following result shows that the value of any contracted MDP lower bounds its non-contracted variant.

**Lemma 2.** *For any $\rho : [H] \times \mathcal{X} \times \mathcal{A} \to [0, 1], \pi \in \Pi_M, h \in [H], k \in [K]$, and $x \in \mathcal{X}$ we have that $\bar{V}_h^{k,\pi}(x; \rho) \leq V_h^{k,\pi}(x)$.*

**Proof.** The proof follows by backward induction on $h \in [H + 1]$. For the base case $h = H + 1$, both values are 0 and the claim holds trivially. Now suppose the claim holds for $h + 1$, then we have that for all $x \in \mathcal{X}$

$$\bar{V}_h^{k,\pi}(x; \rho) = \mathbb{E}_{a \sim \pi(\cdot|x)}\left[\mathbb{E}[\bar{\ell}_h^k(x, a) \mid x, a] + \sum_{x' \in \mathcal{X}} \bar{P}_h(x' \mid x, a)\bar{V}_{h+1}^{k,\pi}(x'; \rho)\right]$$

$$\leq \mathbb{E}_{a \sim \pi(\cdot|x)}\left[\mathbb{E}[\ell_h^k(x, a) \mid x, a] + \sum_{x' \in \mathcal{X}} P_h(x' \mid x, a)V_{h+1}^{k,\pi}(x')\right] = V_h^{k,\pi}(x). \quad \blacksquare$$

Next, for any epoch $e \in [E]$, consider its contracted linear MDP (line 6 in algorithm 1) whose contraction coefficients are $\rho_h^{k_e}(x, a) = \sigma\left(-\beta_w\|\phi(x, a)\|_{(\Lambda_h^{k_e})^{-1}} + \log K\right)$. The following result gives an upper bound on the performance gap between the contracted and non-contracted variants.

**Lemma 3.** *For any $e \in [E]$ and $v \in \mathbb{R}^d$ we have that*

$$(\phi(x_h, a_h) - \bar{\phi}_h^{k_e}(x_h, a_h))^\mathsf{T} v \leq (4\beta_w^2\|\phi(x_h, a_h)\|_{(\Lambda_h^k)^{-1}}^2 + 2K^{-1})|\phi(x_h, a_h)^\mathsf{T} v|.$$

**Proof.** We have that

$$(\phi(x_h, a_h) - \bar{\phi}_h^{k_e}(x_h, a_h))^{\mathsf{T}} v = \sigma(\beta_w \|\phi(x_h, a_h)\|_{(\Lambda_h^{k_e})^{-1}} - \log K) \cdot \phi(x_h, a_h)^{\mathsf{T}} v$$

$$\leq 2(\beta_w^2 \|\phi(x_h, a_h)\|^2_{(\Lambda_h^{k_e})^{-1}} + K^{-1}) |\phi(x_h, a_h)^{\mathsf{T}} v|$$

$$\leq (4\beta_w^2 \|\phi(x_h, a_h)\|^2_{(\Lambda_h^{k})^{-1}} + 2K^{-1}) |\phi(x_h, a_h)^{\mathsf{T}} v|,$$

where the first relation is by the property of the sigmoid $1 - \sigma(x) = \sigma(-x)$, the second is by a simple algebric argument that a quadratic function bounds the sigmoid (lemma 19 in appendix B), and the last relation uses $\det(\Lambda_h^k) \leq 2 \det(\Lambda_h^{k_e})$ by line 4 in algorithm 1 (see lemma 16 in appendix B). ∎

We note that the analogous claim in Sherman et al. [2023a] shows that for all $\pi \in \Pi_M$

$$\mathbb{E}_{P,\pi}[(\phi(x_h, a_h) - \mathbb{1}_{\{x_h \in \mathcal{Z}_h\}}\phi(x_h, a_h))^{\mathsf{T}} v] \leq \Pr(x_h \notin \mathcal{Z}_h) \max_{x,a} |\phi(x, a)^{\mathsf{T}} v|, \tag{6}$$

where $\mathcal{Z}_h$ is an outcome of the reward-free warmup phase and $\Pr(x_h \notin \mathcal{Z}_h) \approx K^{-1/2}$. Summing this over $k \in [K]$ yields a term that scales as $\sqrt{K}$. In contrast, we use a standard bound on elliptical potentials (lemma 15 in appendix B) to get that

$$\sum_{k \in [K]} (4\beta_w^2 \|\phi(x_h^k, a_h^k)\|^2_{(\Lambda_h^k)^{-1}} + 2K^{-1}) \lesssim \log K.$$

This implies that the cost of our contraction is significantly lower than the truncation of Sherman et al. [2023a]. We achieve this reduced cost by using a quadratic (rather than linear) bound on the logistic function. The challenge in our approach is that the above bound only holds for the observed trajectories rather than for all policies as in Sherman et al. [2023a]. In what follows, we overcome this challenge using a novel regret decomposition.

## 5.2 Regret bound

For any epoch $e \in [E]$, let $K_e$ be the set of episodes that it contains, and let $\bar{V}_1^{k,\pi}(x_1; \rho^{k_e})$ denote the value of its contracted MDP as defined above and in line 6 of algorithm 1. We bound the regret as

$$\text{Regret} = \sum_{k \in [K]} V_1^{k,\pi^k}(x_1) - V_1^{k,\pi^\star}(x_1)$$

$$\leq \sum_{e \in [E]} \sum_{k \in K_e} V_1^{k,\pi^k}(x_1) - \bar{V}_1^{k,\pi^\star}(x_1; \rho^{k_e}) \qquad \text{(lemma 2)}$$

$$= \sum_{k \in [K]} V_1^{k,\pi^k}(x_1) - \hat{V}_1^k(x_1) + \sum_{e \in [E]} \sum_{k \in K_e} \hat{V}_1^k(x_1) - \bar{V}_1^{k,\pi^\star}(x_1; \rho^{k_e})$$

$$= \underbrace{\sum_{k \in [K]} V_1^{k,\pi^k}(x_1) - \hat{V}_1^k(x_1)}_{(i) - \text{Bias / Cost of optimism}}$$

$$+ \underbrace{\sum_{e \in [E]} \sum_{h \in [H]} \mathbb{E}_{\bar{P}^{k_e},\pi^\star} \left[ \sum_{k \in K_e} \sum_{a \in \mathcal{A}} \hat{Q}_h^k(x_h, a)(\pi_h^k(a \mid x_h) - \pi_h^\star(a \mid x_h)) \right]}_{(ii) - \text{OMD regret}}$$

$$+ \underbrace{\sum_{e \in [E]} \sum_{k \in K_e} \sum_{h \in [H]} \mathbb{E}_{\bar{P}^{k_e},\pi^\star} \left[ \hat{Q}_h^k(x_h, a_h) - \bar{\phi}_h^{k_e}(x_h, a_h)^{\mathsf{T}}(\theta_h^k + \psi_h \hat{V}_{h+1}^k) \right]}_{(iii) - \text{Optimism}},$$

where the last relation is by the extended value difference lemma (see Shani et al. [2020] and lemma 14 in appendix B). This decomposition is very similar to the standard one for PO algorithms, but with the crucial difference that term $(iii)$ depends on the contracted features $\bar{\phi}_h^{k_e}(x_h, a_h)$ instead of the true features $\phi(x_h, a_h)$. As a by-product, the expectation in terms $(ii)$ and $(iii)$ is taken with respect

to the contracted MDP instead of the true one. The purpose of this modification will be made clear in the proof of optimism (see lemma 4).

In what follows, we bound each term deterministically, conditioned on the following "good event":

$$E_1 = \left\{ \forall k \in [K], h \in [H] : \|\theta_h^k - \widehat{\theta}_h^k\|_{\Lambda_h^k} \leq \beta_r \right\}; \tag{7}$$

$$E_2 = \left\{ k \in [K], h \in [H] : \|(\psi_h - \widehat{\psi}_h^k)\hat{V}_{h+1}^k\|_{\Lambda_h^k} \leq \beta_p, \|\hat{Q}_{h+1}^k\|_\infty \leq 2H \right\}. \tag{8}$$

$E_1$ and $E_2$ are error bounds on the loss and dynamics estimation, respectively. In the full feedback setting, $E_1$ holds trivially with $\beta_r = 0$. In the bandit setting, it holds with high probability with $\beta_r = O(\sqrt{d \log(KH/\delta)})$ by well-established bounds for regularized least-squares estimation [Abbasi-Yadkori et al., 2011]. Showing that $E_2$ holds with high probability follows similarly to Sherman et al. [2023a], again using least-squares arguments but also using the contraction to ensure that $\hat{Q}_h^k$ are bounded (see sketch at the end of this section and lemma 6 in appendix A for full details), specifically $\beta_p = O(Hd\sqrt{\log(KH/\delta)})$. The proof of theorem 1 is concluded by bounding each of the terms in the regret decomposition, summing over $k \in [K]$ and using a standard bound on elliptical potentials (lemma 15 in appendix B). Term $(ii)$ is bounded using a standard Online Mirror Descent (OMD) argument (lemma 7 in appendix A).

**Optimism and its cost.** The following lemmas bound terms $(iii)$ and $(i)$, respectively.

**Lemma 4 (Optimism).** *Suppose that eqs. (7) and (8) hold, then*

$$\hat{Q}_h^k(x,a) - \bar{\phi}_h^{k_e}(x,a)^\mathsf{T}(\theta_h + \psi_h \hat{V}_{h+1}^k) \leq 0 \quad, \forall h \in [H], k \in [K], x \in \mathcal{X}, a \in \mathcal{A}.$$

**Proof.** We have that

$$\hat{Q}_h^k(x,a) - \bar{\phi}_h^{k_e}(x,a)^\mathsf{T}(\theta_h + \psi_h \hat{V}_{h+1}^k) = \bar{\phi}_h^{k_e}(x,a)^\mathsf{T}(\widehat{\theta}_h^k - \theta_h + (\widehat{\psi}_h^k - \psi_h)\hat{V}_{h+1}^k)$$
$$- \beta_b \|\bar{\phi}_h^{k_e}(x,a)\|_{(\Lambda_h^{k_e})^{-1}}$$
$$\leq (\beta_r + \beta_p)\|\bar{\phi}_h^{k_e}(x,a)\|_{\Lambda_h^{k-1}} - \beta_b \|\bar{\phi}_h^{k_e}(x,a)\|_{(\Lambda_h^{k_e})^{-1}}$$
$$\leq (\beta_r + \beta_p - \beta_b)\|\bar{\phi}_h^{k_e}(x,a)\|_{(\Lambda_h^{k_e})^{-1}} = 0,$$

where the first relation is by definition of $\hat{Q}_h^k$ (eq. (3) in algorithm 1), the second relation is by eqs. (7) and (8) together with Cauchy-Schwarz, the third relation follows since $\Lambda_h^{k_e} \preceq \Lambda_h^k$ and the last one is by our choice $\beta_b = \beta_r + \beta_p$ (see theorem 9 in appendix A for hyper-parameter choices). ∎

Notice that the standard PO decomposition would have required that we bound the non-contracted expression $\mathbb{E}_{P,\pi^\star}[\hat{Q}_h^k(x,a) - \phi(x,a)^\mathsf{T}(\theta_h^k + \psi_h \hat{V}_{h+1}^k)]$. In Sherman et al. [2023a] the gap between this argument and that of lemma 4 can be bounded using eq. (6). However, the equivalent argument for our contraction is lemma 3, which is bounded only for $\pi^k$ and not for any policy $\pi \in \Pi_M$.

**Lemma 5 (Cost of optimism).** *Suppose that eqs. (7) and (8) hold, then for every $k \in [K]$*

$$V_1^{k,\pi^k}(x_1) - \hat{V}_1^k(x_1) \leq 3(\beta_r + \beta_p)\mathbb{E}_{P,\pi^k}\left[\sum_{h \in [H]} \|\phi(x_h,a_h)\|_{(\Lambda_h^k)^{-1}}\right]$$

$$+ 16H\beta_w^2 \mathbb{E}_{P,\pi^k}\left[\sum_{h \in [H]} \|\phi(x_h,a_h)\|_{(\Lambda_h^k)^{-1}}^2\right] + 16H^2 K^{-1}.$$

**Proof.** First, by lemma 14 in appendix B, a value difference lemma by Shani et al. [2020],

$$V_1^{k,\pi^k}(x_1) - \hat{V}_1^k(x_1) = \mathbb{E}_{P,\pi^k}\left[\sum_{h \in [H]} \phi(x_h,a_h)^\mathsf{T}\left(\theta_h + \psi_h \hat{V}_{h+1}^k\right) - \hat{Q}_k^k(x_h,a_h)\right].$$

Now, using lemma 3 with $v = \theta_h^k + \psi_h \hat{V}_{h+1}^k$ we have that $|\phi(x,a)^\mathsf{T}v| \leq 4H$ (by eq. (8)) and thus

$$[\phi(x_h,a_h) - \bar{\phi}_h^{k_e}(x_h,a_h)]^\mathsf{T}\left(\theta_h + \psi_h \hat{V}_{h+1}^k\right) \leq 16H\beta_w^2 \|\phi(x_h,a_h)\|_{(\Lambda_h^k)^{-1}}^2 + 16H^2 K^{-1}.$$

We can thus conclude the proof using standard arguments to show that

$$\bar{\phi}_h^{k_e}(x_h,a_h)^{\mathsf{T}}\left(\theta_h + \psi_h \hat{V}_{h+1}^k\right) - \hat{Q}_k^k(x_h,a_h)$$

$$= \bar{\phi}_h^{k_e}(x_h,a_h)^{\mathsf{T}}\left(\theta_h^k - \widehat{\theta}_h^k + (\psi_h - \widehat{\psi}_h^k)\hat{V}_{h+1}^k\right) + \beta_b\|\bar{\phi}_h^{k_e}(x_h,a_h)\|_{(\Lambda_h^{k_e})^{-1}} \qquad \text{(eq. (3))}$$

$$\leq (\beta_r + \beta_p)\|\bar{\phi}_h^{k_e}(x_h,a_h)\|_{(\Lambda_h^k)^{-1}} + \beta_b\|\bar{\phi}_h^{k_e}(x_h,a_h)\|_{(\Lambda_h^{k_e})^{-1}}$$
$$\text{(Cauchy-Schwarz, eqs. (7) and (8))}$$

$$\leq 3(\beta_r + \beta_p)\|\bar{\phi}_h^{k_e}(x_h,a_h)\|_{(\Lambda_h^k)^{-1}} \qquad\qquad (\det(\Lambda_h^k) \leq 2\det(\Lambda_h^{k_e}), \beta_b = \beta_r + \beta_p)$$

$$\leq 3(\beta_r + \beta_p)\|\phi(x_h,a_h)\|_{(\Lambda_h^k)^{-1}}, \qquad\qquad (\sigma(x) \in [0,1], \forall x \in \mathbb{R})$$

as desired. $\blacksquare$

**Bounding the Q-values (proof sketch).** The following are the main ideas in showing that $E_2$ (eq. (8)) holds with high probability. First, we define appropriate value classes $\widehat{\mathcal{V}}_h$ that contain all value functions $V_h$ of the form in eq. (4) whose underlying $Q_h$ function (eq. (3)) satisfies $\|Q_h\|_\infty \leq 2(H + 1 - h)$. Because both the bonus and contraction operator are kept fixed during each epoch, the log covering number of this class is logarithmic in $K$ (similarly to Sherman et al. [2023a]). Thus, we can use standard least squares arguments (lemma 22) to show that with high probability $\|(\psi_h - \widehat{\psi}_h^k)V\|_{\Lambda_h^k} \leq \beta_p$ for all $k \in [K], h \in [H], V \in \widehat{\mathcal{V}}_h$. The proof is concluded by showing that $\|\hat{Q}_h^k\| \leq \beta_{Q,h} = 2(H + 1 - h)$, and thus $\hat{V}_h^k \in \widehat{\mathcal{V}}_h$, which implies that eq. (8) holds. We prove this by backward induction on $h \in [H + 1]$.

The base case $h = H + 1$ is satisfied because, by definition, $\hat{Q}_{H+1}^k = 0$. Now, suppose the claim holds for $h + 1$ and we show it also holds for $h$. Recalling the definition of $\hat{Q}$ in eq. (3), we have that

$$|\hat{Q}_h^k(x,a)| = |\bar{\phi}_h^{k_e}(x,a)^{\mathsf{T}}(\widehat{\theta}_h^k + \widehat{\psi}_h^k\hat{V}_{h+1}^k) - \beta_b\|\bar{\phi}_h^{k_e}(x,a)\|_{(\Lambda_h^{k_e})^{-1}}|$$

$$\leq |\bar{\phi}_h^{k_e}(x,a)^{\mathsf{T}}(\theta_h + (\widehat{\theta}_h^k - \theta_h) + (\widehat{\psi}_h^k - \psi_h)\hat{V}_{h+1}^k + \psi_h\hat{V}_{h+1}^k)| + \beta_b\|\bar{\phi}_h^{k_e}(x,a)\|_{(\Lambda_h^{k_e})^{-1}}$$

$$\leq 1 + \|\hat{V}_{h+1}^k\|_\infty + \|\bar{\phi}_h^{k_e}(x,a)\|_{(\Lambda_h^{k_e})^{-1}}\left[\|\widehat{\theta}_h^k - \theta_h\|_{\Lambda_h^k} + \|(\widehat{\psi}_h^k - \psi_h)\hat{V}_{h+1}^k\|_{\Lambda_h^k} + \beta_b\right],$$

where the last inequality also used the triangle and Cauchy-Schwarz inequalities, and that $\Lambda_h^{k_e} \preceq \Lambda_h^k$. By the induction hypothesis, $\|\hat{Q}_{h+1}^k\|_\infty, \|\hat{V}_{h+1}^k\|_\infty \leq \beta_{Q,h+1}$ and thus $\hat{V}_{h+1}^k \in \widehat{\mathcal{V}}_{h+1}$. Combining with $E_1$ (eq. (7)) and plugging into the above we get that

$$|\hat{Q}_h^k(x,a)| \leq 1 + \beta_{Q,h+1} + (\beta_r + \beta_{p,h} + \beta_b)\|\bar{\phi}_h^{k_e}(x,a)\|_{(\Lambda_h^{k_e})^{-1}}.$$

Now, using a technical algebraic argument (lemma 18), we show that

$$\|\bar{\phi}_h^{k_e}(x,a)\|_{(\Lambda_h^{k_e})^{-1}} \leq \max_{y \geq 0}[y \cdot \sigma(-\beta_w y + \log K)] \leq 2\beta_w^{-1}\log(eK).$$

Finally, plugging this into the above and choosing $\beta_w \geq 2(\beta_r + \beta_{p,h} + \beta_b)\log(eK)$, we get

$$|\hat{Q}_h^k(x,a)| \leq 1 + \beta_{Q,h+1} + 2\beta_w^{-1}(\beta_r + \beta_{p,h} + \beta_b)\log(eK) \leq 2 + \beta_{Q,h+1} = \beta_{Q,h},$$

concluding the induction.

# 6 Conclusions

In this paper we presented a simple and efficient contraction mechanism for policy optimization in linear MDPs, yielding an overall algorithm with improved regret guarantees under both stochastic (bandit feedback) and adversarial (full feedback) losses. We note that, in the stochastic setting, there are value iteration based methods (He et al. [2023]) that use variance reduction techniques to achieve better regret bounds. We conjecture that such techniques could be applicable to PO, however, this is highly non-trivial and thus left for future research. Finally, regarding practical implementations, we note that our bonuses and contraction technique are computationally feasible, especially compared to the reward-free warmup phase in Sherman et al. [2023a]. Nonetheless, it remains open whether our techniques could be applied heuristically to drive exploration in practical deep RL methods. In particular, it would be interesting to examine the necessity of the contraction mechanism. These are challenging questions on exploration in deep RL that we leave for future research.

## Acknowledgments and Disclosure of Funding

This project has received funding from the European Research Council (ERC) under the European Union's Horizon 2020 research and innovation program (grant agreement No. 101078075). Views and opinions expressed are however those of the author(s) only and do not necessarily reflect those of the European Union or the European Research Council. Neither the European Union nor the granting authority can be held responsible for them. This work received additional support from the Israel Science Foundation (ISF, grant number 2549/19), the Len Blavatnik and the Blavatnik Family Foundation, and the Israeli VATAT data science scholarship.

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

# A  Analysis

We begin by defining a so-called "good event", followed by optimism, cost of optimism, and Policy Optimization cost. We conclude with the proof of theorem 9.

**Good event.**  We define the following good event $E_g = \bigcap_{i=1}^{3} E_i$, over which the regret is deterministically bounded:

$$E_1 = \left\{ \forall k \in [K], h \in [H] : \|\theta_h^k - \widehat{\theta}_h^k\|_{\Lambda_h^k} \leq \beta_r \right\}; \tag{eq. (7)}$$

$$E_2 = \left\{ k \in [K], h \in [H] : \|(\psi_h - \widehat{\psi}_h^k)\hat{V}_{h+1}^k\|_{\Lambda_h^k} \leq \beta_p, \|\hat{Q}_{h+1}^k\|_\infty \leq \beta_Q \right\}; \tag{eq. (8)}$$

$$E_3 = \left\{ \sum_{k \in [K]} \mathbb{E}_{P,\pi^k}[Y_k] \leq \sum_{k \in [K]} 2Y_k + 4H(3(\beta_r + \beta_p) + 4\beta_Q\beta_w^2) \log \frac{6}{\delta} \right\}. \tag{9}$$

where $Y_k = \sum_{h \in [H]} 3(\beta_r + \beta_p)\|\phi(x_h, a_h)\|_{(\Lambda_h^k)^{-1}} + 4\beta_Q\beta_w^2\|\phi(x_h, a_h)\|_{(\Lambda_h^k)^{-1}}^2$.

**Lemma 6 (Good event).** *Consider the parameter setting of theorem 9. If $\eta_o \leq 1, \beta_w^2 \leq K/(32Hd)$ then $\Pr[E_g] \geq 1 - \delta$.*

Proof in appendix A.1.

**Policy online mirror descent.**  We use standard online mirror descent arguments to bound the local regret in each state.

**Lemma 7 (OMD).** *Suppose that the good event $E_g$ holds (eqs. (7), (8) and (9)) and set $\eta_o \leq 1/\beta_Q$, then*

$$\sum_{k \in K_e} \sum_{a \in \mathcal{A}} \hat{Q}_h^k(x, a)(\pi_h^\star(a \mid x) - \pi_h^k(a \mid x)) \leq \frac{\log|\mathcal{A}|}{\eta_o} + \eta_o \sum_{k \in K_e} \beta_Q^2 \quad, \forall e \in [E], h \in [H], x \in \mathcal{X}.$$

**Proof.** Notice that the policy $\pi^k$ is reset at the beginning of every epoch. Then, the lemma follows directly by lemma 13 with $y_t(a) = -\hat{Q}_h^k(x, a), x_t(a) = \pi_h^k(a \mid x)$ and noting that $|\hat{Q}_h^k(x, a)| \leq \beta_Q$ by eq. (8). ∎

**Epoch schedule.**  The algorithm operates in epochs. At the beginning of each epoch, the policy is reset to be uniformly random. We denote the total number of epochs by $E$, the first episode within epoch $e$ by $k_e$, and the set of episodes within epoch $e$ by $K_e$. The following lemma bounds the number of epochs.

**Lemma 8.** *The number of epochs $E$ is bounded by $(3/2)dH \log(2K)$.*

**Proof.** Let $\mathcal{T}_h = \{e_h^1, e_h^2, \dots\}$ be the epochs where the condition $\det(\Lambda_h^k) \geq 2\det(\Lambda_h^{k_e})$ was triggered in line 4 of algorithm 1. Then we have that

$$\det(\Lambda_h^{k_e}) \geq \begin{cases} 2\det(\Lambda_h^{k_{e-1}}) & , e \in \mathcal{T}_h \\ \det(\Lambda_h^{k_{e-1}}) & , \text{otherwise.} \end{cases}$$

Unrolling this relation, we get that

$$\det(\Lambda_h^K) \geq 2^{|\mathcal{T}_h|-1} \det I = 2^{|\mathcal{T}_h|-1},$$

and changing sides, and taking the logarithm we get that

$$\begin{aligned} |\mathcal{T}_h| &\leq 1 + \log_2 \det\left(\Lambda_h^K\right) \\ &\leq 1 + d\log_2\|\Lambda_h^K\| && (\det(A) \leq \|A\|^d) \\ &\leq 1 + d\log_2\left(1 + \sum_{k=1}^{K-1}\|\phi_h^k\|^2\right) && (\text{triangle inequality}) \\ &\leq 1 + d\log_2 K && (\|\phi_h^k\| \leq 1) \\ &\leq (3/2)d\log 2K. \end{aligned}$$

We conclude that

$$E = |(\cup_{h\in[H]}\mathcal{T}_h)| \le \sum_{h\in[H]} |\mathcal{T}_h| \le (3/2)dH\log(2K). \qquad \blacksquare$$

**Regret bound.**

**Theorem 9.** *Suppose that we run algorithm 1 with parameters*

$$\eta_o = \sqrt{\frac{3dH\log(2K)\log|\mathcal{A}|}{K\beta_Q^2}}, \beta_b = \beta_r + \beta_p, \beta_w = 4(\beta_r + \beta_p)\log(eK),$$

*where $\beta_r = 2\sqrt{2d\log(6KH/\delta)}, \beta_p = 28Hd\sqrt{\log(10K^5H/\delta)}, \beta_Q = 2H$. Then with probability at least $1 - \delta$ we incur regret at most*

$$\text{Regret} \le 264\sqrt{Kd^3H^4\log(2K)\log(10K^5H/\delta)} + 8\sqrt{KdH^5\log(2K)\log|\mathcal{A}|}$$
$$+ 64H^2d\max\{\beta_w^2,\log|\mathcal{A}|\}\log\frac{12K}{\delta}$$
$$= O(\sqrt{Kd^3H^4\log(K)\log(KH/\delta)} + \sqrt{KdH^5\log(K)\log|\mathcal{A}|}).$$

**Proof.** First, if $\beta_w^2 > K/(32Hd)$ or $\eta \ge 1/\beta_Q$ then

$$\text{Regret} \le KH \le 32H^2d\max\{\beta_w^2,\log|\mathcal{A}|\}\log(2K),$$

and the proof is concluded. Otherwise, if $\beta_w^2 \le K/(32Hd)$ then suppose that the good event $E_g$ holds (eqs. (7), (8) and (9)). By lemma 6, this holds with probability at least $1 - \delta$. For any epoch $e \in [E]$, let $K_e$ be the set of episodes that it contains, and let $\bar{V}_1^{k,\pi}(x_1; \rho^{k_e})$ denote the value of its contracted MDP as defined in section 5.1 and line 6 of algorithm 1. We bound the regret as

$$\text{Regret} = \sum_{k\in[K]} V_1^{k,\pi^k}(x_1) - V_1^{k,\pi^\star}(x_1)$$

$$\le \sum_{e\in[E]}\sum_{k\in K_e} V_1^{k,\pi^k}(x_1) - \bar{V}_1^{k,\pi^\star}(x_1; \rho^{k_e}) \qquad \text{(lemma 2)}$$

$$= \sum_{k\in[K]} V_1^{k,\pi^k}(x_1) - \hat{V}_1^k(x_1) + \sum_{e\in[E]}\sum_{k\in K_e} \hat{V}_1^k(x_1) - \bar{V}_1^{k,\pi^\star}(x_1; \rho^{k_e})$$

$$= \underbrace{\sum_{k\in[K]} V_1^{k,\pi^k}(x_1) - \hat{V}_1^k(x_1)}_{(i)-\text{Bias / Cost of optimism}}$$

$$+ \underbrace{\sum_{e\in[E]}\sum_{h\in[H]} \mathbb{E}_{\bar{P}^{k_e},\pi^\star}\left[\sum_{k\in K_e}\sum_{a\in\mathcal{A}} \hat{Q}_h^k(x_h,a)(\pi_h^k(a\mid x_h) - \pi_h^\star(a\mid x_h))\right]}_{(ii)-\text{OMD regret}}$$

$$+ \underbrace{\sum_{e\in[E]}\sum_{k\in K_e}\sum_{h\in[H]} \mathbb{E}_{\bar{P}^{k_e},\pi^\star}\left[\hat{Q}_h^k(x_h,a_h) - \bar{\phi}_h^{k_e}(x_h,a_h)^\mathsf{T}(\theta_h^k + \psi_h\hat{V}_{h+1}^k)\right]}_{(iii)-\text{Optimism}},$$

where the last relation is by the extended value difference lemma (see Shani et al. [2020] and lemma 14 in appendix B).

For term $(i)$, we use lemma 5 as follows.

$$(i) \leq \sum_{k \in [K]} \mathbb{E}_{P,\pi^k} \left[ \sum_{h \in [H]} 3(\beta_r + \beta_p)\|\phi(x_h, a_h)\|_{(\Lambda_h^k)^{-1}} + 8\beta_Q \beta_w^2 \|\phi(x_h, a_h)\|_{(\Lambda_h^k)^{-1}}^2 \right] + 8H\beta_Q$$

$$\leq \sum_{k \in [K]} \left[ \sum_{h \in [H]} 6(\beta_r + \beta_p)\|\phi(x_h^k, a_h^k)\|_{(\Lambda_h^k)^{-1}} + 16\beta_Q \beta_w^2 \|\phi(x_h^k, a_h^k)\|_{(\Lambda_h^k)^{-1}}^2 \right] + 20H\beta_Q \beta_w^2 \log \frac{6}{\delta}$$

$$\text{(eq. (9), } \beta_w \geq 4(\beta_r + \beta_p) \geq 32)$$

$$\leq 6(\beta_r + \beta_p)H\sqrt{2Kd\log(2K)} + 32\beta_Q \beta_w^2 Hd\log(2K) + 20H\beta_Q \beta_w^2 \log \frac{6}{\delta} \qquad \text{(lemma 15)}$$

$$\leq 6(\beta_r + \beta_p)H\sqrt{2Kd\log(2K)} + 32Hd\beta_Q \beta_w^2 \log \frac{12K}{\delta}.$$

By lemmas 7 and 8 (with our choice of $\eta_o$) we have

$$(ii) \leq \sum_{h \in [H]} \sum_{e \in [E]} \mathbb{E}_{\bar{P}^{k_e}, \pi^\star} \left[ \frac{\log A}{\eta_o} + \eta_o \sum_{k \in K_e} \beta_Q^2 \right] \leq 4H\beta_Q \sqrt{KdH \log(2K) \log|\mathcal{A}|}.$$

By lemma 4 $(iii) \leq 0$. Putting all bounds together, we get that

$$\text{Regret} \leq 6(\beta_r + \beta_p)H\sqrt{2Kd\log(2K)} + 32Hd\beta_Q \beta_w^2 \log \frac{12K}{\delta} + 4H\beta_Q \sqrt{KdH \log(2K) \log|\mathcal{A}|}$$

$$\leq 264\sqrt{Kd^3H^4 \log(2K)\log(10K^5H/\delta)} + 8\sqrt{KdH^5 \log(2K)\log|\mathcal{A}|} + 64H^2 d\beta_w^2 \log \frac{12K}{\delta}$$

$$= O(\sqrt{Kd^3H^4 \log(K)\log(KH/\delta)} + \sqrt{KdH^5 \log(K)\log|\mathcal{A}|}). \qquad \blacksquare$$

## A.1 Proofs of good event

We begin by defining function classes and properties necessary for the uniform convergence arguments over the value functions. We then proceed to define a proxy good event, whose high probability occurrence is straightforward to prove. We then show that the proxy event implies the desired good event.

**Value and policy classes.** We define the following class of restricted Q-functions:

$$\widehat{\mathcal{Q}}(C_\beta, C_w, C_Q)$$
$$= \left\{ \hat{Q}(\cdot, \cdot; \beta, w, \Lambda, \mathcal{Z}) \mid \beta \in [0, C_\beta], \|w\| \leq C_w, (2K)^{-1}I \preceq \Lambda \preceq I, \|\hat{Q}(\cdot, \cdot; w, \Lambda, \mathcal{Z})\|_\infty \leq C_Q \right\},$$

where $\hat{Q}(x, a; \beta, w, \Lambda) = [w^\mathsf{T}\phi(x, a) - \beta\|\phi(x, a)\|_\Lambda] \cdot \sigma(-\beta_w \|\phi(x, a)\|_\Lambda + \log K)$. Next, we define the following class of soft-max policies:

$$\Pi(C_\beta, C_w) = \left\{ \pi(\cdot \mid \cdot; \hat{Q}) \mid \hat{Q} \in \widehat{\mathcal{Q}}(C_\beta, C_w, \infty) \right\},$$

where $\pi(a \mid x; \hat{Q}) = \frac{\exp(\hat{Q}(x,a))}{\sum_{a' \in \mathcal{A}} \exp(\hat{Q}(x,a'))}$. Finally, we define the following class of restricted value functions:

$$\widehat{\mathcal{V}}(C_\beta, C_w, C_Q) = \left\{ \hat{V}(\cdot; \pi, \hat{Q}) \mid \pi \in \Pi(C_\beta K, C_w K, C_Q), \hat{Q} \in \widehat{\mathcal{Q}}(C_\beta, C_w, C_Q) \right\}, \qquad (10)$$

where $\hat{V}(x; \pi, \hat{Q}) = \sum_{a \in \mathcal{A}} \pi(a \mid x)\hat{Q}(x, a)$. The following lemma provides the bound on the covering number of the value function class defined above.

**Lemma 10.** *For any $\epsilon, C_w > 0, C_\beta, C_Q \geq 1$, we have*

$$\log \mathcal{N}_\epsilon\left(\widehat{\mathcal{V}}(C_\beta, C_w, C_Q)\right) \leq 6d^2 \log(1 + 4(\sqrt{192K^3}C_Q C_\beta \beta_w)(KC_\beta + KC_w + \sqrt{d})/\epsilon),$$

*where $\mathcal{N}_\epsilon$ is the covering number of a class in supremum distance.*

**Proof.** We begin by showing that the class of $Q$ function is Lipschitz in its parameters. For ease of notation, denote $y = \phi(x, a)$. Then

$$\|\nabla_\beta Q(x, a; \beta, w, \Lambda)\| = \|y\|_\Lambda \cdot \sigma(-\beta_w\|y\|_\Lambda + \log K) \leq 1 \quad (\sigma(\cdot) \in [0, 1], \|y\| \leq 1, \Lambda \preceq I)$$

$$\|\nabla_\theta \hat{Q}(x, a; \beta, w, \Lambda)\| = \|y \cdot \sigma(-\beta_w\|y\|_\Lambda + \log K)\| \leq 1 \quad (\sigma(\cdot) \in [0, 1], \|y\| \leq 1)$$

$$\begin{aligned}
|Q(x, a; &\beta, w, \Lambda) - Q(x, a; \beta, w, \Lambda')| \\
&\leq \beta|\|y\|_\Lambda - \|y\|_{\Lambda'}| \cdot \sigma(-\beta_w\|y\|_\Lambda + \log K) \\
&\quad + \beta\|y\|_{\Lambda'}|\sigma(-\beta_w\|y\|_\Lambda + \log K) - \sigma(-\beta_w\|y\|_{\Lambda'} + \log K)| \\
&\leq \beta\|(\Lambda^{1/2} - (\Lambda')^{1/2})y\| + \beta\beta_w\|y\|_{\Lambda'}\|(\Lambda^{1/2} - (\Lambda')^{1/2})y\| \\
&\hspace{4cm} (\|\cdot\|, \sigma(\cdot) \text{ 1-Lipschitz}, \sigma \in [0, 1]) \\
&\leq 2\beta\beta_w\|\Lambda^{1/2} - (\Lambda')^{1/2}\| \hspace{2cm} (\|y\| \leq 1, \Lambda \preceq I, \beta_w \geq 1) \\
&\leq \sqrt{2K}\beta\beta_w\|\Lambda - \Lambda'\| \hspace{2cm} (\text{lemma 17}, \Lambda, \Lambda' \succeq (2K)^{-1}I) \\
&\leq \sqrt{2K}\beta\beta_w\|\Lambda - \Lambda'\|_F. \hspace{2cm} (\|\cdot\| \leq \|\cdot\|_F)
\end{aligned}$$

We thus have that for any such $y$

$$\begin{aligned}
|Q(x, a; \beta, w, &\Lambda) - Q(x, a; \beta', w', \Lambda')| \\
&\leq |Q(x, a; \beta, w, \Lambda) - Q(x, a; \beta', w, \Lambda)| + |Q(x, a; \beta', w, \Lambda) - Q(x, a; \beta', w', \Lambda)| \\
&\quad + |Q(x, a; \beta', w', \Lambda) - Q(x, a; \beta', w', \Lambda')| \\
&\leq |\beta - \beta'| + \|w - w'\| + \sqrt{2K}\beta\beta_w\|\Lambda - \Lambda'\|_F \\
&\leq \sqrt{3(\|w - w'\|^2 + |\beta - \beta'|^2 + (\sqrt{2K}\beta\beta_w)^2\|\Lambda - \Lambda'\|_F^2)} \\
&\leq \max\{3, \sqrt{6K}\beta\beta_w\}\sqrt{(\|w - w'\|^2 + |\beta - \beta'|^2 + \|\Lambda - \Lambda'\|_F^2)} \\
&= \max\{3, \sqrt{6K}\beta\beta_w\}\|(\beta, w, \Lambda) - (\beta', w', \Lambda')\|,
\end{aligned}$$

where $(\beta, w, \Lambda)$ is a vectorization of the parameters. Assuming that $C_\beta \geq 1$, we conclude that $\hat{\mathcal{Q}}(C_\beta, C_w, C_Q)$ is $\sqrt{6K}C_\beta\beta_w$−Lipschitz in supremum norm, i.e.,

$$\|\hat{Q}(\cdot, \cdot; \beta, w, \Lambda) - \hat{Q}'(\cdot, \cdot; \beta', w', \Lambda')\|_\infty \leq \sqrt{6K}C_\beta\beta_w\|(\beta, w, \Lambda) - (\beta', w', \Lambda')\|.$$

Next, notice that our policy class $\Pi(C_\beta K, C_w K)$ is a soft-max over the Q functions thus fitting Lemma 12 of Sherman et al. [2023a]. We conclude that the policy class is $\sqrt{24K^3}C_\beta\beta_w$−Lipschitz, in $\ell_1$−norm, i.e.,

$$\|\pi(\cdot \mid x; \beta, w, \Lambda) - \pi(\cdot \mid x; \beta', w', \Lambda')\|_1 \leq \sqrt{24K^3}C_\beta\beta_w\|(\beta, w, \Lambda) - (\beta', w', \Lambda')\|.$$

Now, let $V, V' \in \hat{\mathcal{V}}(C_\beta, C_w, C_Q)$ and $\theta = (\beta_1, w_1, \Lambda_1, \beta_2, w_2, \Lambda_2), \theta' = (\beta_1', w_1', \Lambda_1', \beta_2', w_2', \Lambda_2) \in \mathbb{R}^{2(1+d+d^2)}$ be their respective parameters. We have that for all $x \in \mathcal{X}$

$$|V(x; \pi, \hat{Q}) - V(x; \pi', \hat{Q}')| \leq \underbrace{|V(x; \pi, \hat{Q}) - V(x; \pi, \hat{Q}')|}_{(i)} + \underbrace{|V(x; \pi, \hat{Q}') - V(x; \pi', \hat{Q}')|}_{(ii)}.$$

For the first term

$$\begin{aligned}
(i) &= \left|\sum_{a \in \mathcal{A}} \pi(a \mid x)(\hat{Q}(x, a; \beta_2, w_2, \Lambda_2) - \hat{Q}(x, a; \beta_2', w_2', \Lambda_2'))\right| \\
&\leq \sum_{a \in \mathcal{A}} \pi(a \mid x)\left|\hat{Q}(x, a; \beta_2, w_2, \Lambda_2) - \hat{Q}(x, a; \beta_2', w_2', \Lambda_2')\right| \hspace{1cm} (\text{triangle inequality}) \\
&\leq \sqrt{6K}C_\beta\beta_w\|(\beta_2, w_2, \Lambda_2) - (\beta_2', w_2', \Lambda_2')\|. \quad (\hat{Q} \text{ is } \sqrt{6K}C_\beta\beta_w\text{-Lipschitz, Cauchy-Schwarz})
\end{aligned}$$

For the second term

$$\begin{aligned}
(ii) &= \left|\sum_{a \in \mathcal{A}} \hat{Q}'(x, a)(\pi(a \mid x) - \pi'(a \mid x))\right| \leq C_Q\|\pi(\cdot \mid x) - \pi(\cdot \mid x)\|_1 \\
&\leq \sqrt{96K^3}C_Q C_\beta\beta_w\|(\beta_1, w_1, \Lambda_1) - (\beta_1', w_1', \Lambda_1')\|,
\end{aligned}$$

where the first transition used that $\|Q\|_\infty \le C_Q$ for all $Q \in \widehat{\mathcal{Q}}(C_\beta, C_w, C_Q)$ and the second used the Lipschitz property of the policy class shown above. Combining the terms and assuming that $C_Q \ge 1$ we get that

$$
\begin{aligned}
|V(x; \pi, \hat{Q}) - V(x; \pi', \hat{Q}')| &\le \sqrt{96K^3} C_Q C_\beta \beta_w \|(\beta_1, w_1, \Lambda_1) - (\beta_1', w_1', \Lambda_1')\| \\
&\quad + \sqrt{96K^3} C_Q C_\beta \beta_w \|(\beta_2, w_2, \Lambda_2) - (\beta_2', w_2', \Lambda_2')\| \\
&\le \sqrt{192K^3} C_Q C_\beta \beta_w \|\theta - \theta'\|,
\end{aligned}
$$

implying that $\widehat{\mathcal{V}}(C_\beta, C_w, C_Q)$ is $\sqrt{192K^3} C_Q C_\beta \beta_w$−Lipschitz in supremum norm. Finally, notice that

$$
\|\theta\| \le |\beta_1| + |\beta_2| + \|w_1\| + \|w_2\| + \|\Lambda_1\|_F + \|\Lambda_2\|_F \le 2KC_\beta + 2KC_w + 2\sqrt{d},
$$

and applying lemma 24 concludes the proof. ∎

**Proxy good event.** We define a proxy good event $\bar{E}_g = E_1 \cap \bar{E}_2 \cap E_3$ where

$$
\bar{E}_2 = \left\{ k \in [K], h \in [H], V \in \widehat{\mathcal{V}}(\beta_r + \beta_p, 2\beta_Q K, \beta_{Q,h+1}) : \|(\psi_h - \widehat{\psi}_h^k)V\|_{\Lambda_h^k} \le \beta_p \right\}, \tag{11}
$$

where $\beta_{Q,h} = 2(H + 1 - h), h \in [H + 1]$. Then we have the following result.

**Lemma 11 (Proxy good event).** *Consider the parameter setting of lemma 6. Then* $\Pr[\bar{E}_g] \ge 1 - \delta$.

**Proof.** First, by lemma 21 and our choice of parameters, $E_1$ (eq. (7)) holds with probability at least $1 - \delta/3$. Next, applying lemmas 10 and 22, we get that with probability at least $1 - \delta/3$ simultaneously for all $k \in [K], h \in [H], V \in \widehat{\mathcal{V}}(\beta_r + \beta_p, 2\beta_Q K, \beta_{Q,h+1})$

$$
\begin{aligned}
&\|(\psi_h - \widehat{\psi}_h^k)V\|_{\Lambda_h^k} \\
&\le 4\beta_{Q,h+1} \sqrt{d\log(2K) + 2\log(6H/\delta) + 12d^2\log(1 + 8K(\sqrt{192K^3}C_\beta\beta_w)(KC_\beta + KC_w + 1))} \\
&\le 4\beta_Q \sqrt{d\log(2K) + 2\log(6H/\delta) + 12d^2\log(1 + 2K(\sqrt{192K^3}K/(32Hd))(\tfrac{1}{4}K\sqrt{K/(32Hd)} + 2\beta_Q K^2 + 1))} \\
&\le 4\beta_Q \sqrt{d\log(2K) + 2\log(6H/\delta) + 12d^2\log(7K^{9/2})} \\
&\le 4\beta_Q d \sqrt{12\log(10K^5 H/\delta)} \\
&\le 28Hd\sqrt{\log(10K^5 H/\delta)} \\
&= \beta_p,
\end{aligned}
$$

implying $\bar{E}_2$ (eq. (11)). Finally, notice that $\|\phi_h^k\|_{(\Lambda_h^k)^{-1}} \le 1$, thus $0 \le Y_k \le H(3(\beta_r + \beta_p) + 4\beta_Q \beta_w^2)$. Using lemma 20, a Bernstein-type inequality for martingales, we conclude that $E_3$ (eq. (9)) holds with probability at least $1 - \delta/3$. ∎

**The good event.** The following results show that the proxy good event implies the good event.

**Lemma 12.** *Suppose that $\bar{E}_g$ holds. If $\pi_h^k \in \Pi(K(\beta_r + \beta_p), 2\beta_Q K^2)$ for all $h \in [H]$ then $\hat{Q}_h^k \in \widehat{\mathcal{Q}}(\beta_r + \beta_p, 2\beta_Q K, \beta_{Q,h}), \hat{V}_h^k \in \widehat{\mathcal{V}}(\beta_r + \beta_p, 2\beta_Q K, \beta_{Q,h})$ for all $h \in [H + 1]$.*

**Proof.** We show that the claim holds by backward induction on $h \in [H + 1]$.
**Base case $h = H + 1$:** Since $\hat{V}_{H+1}^k = 0$ it is also implied that $\hat{Q}_{H+1}^k = 0$. Because $\beta, w = 0 \in \widehat{\mathcal{Q}}(\beta_r + \beta_p, 2\beta_Q K, \beta_{Q,H+1} = 0)$ we have that $\hat{Q}_{H+1}^k \in \widehat{\mathcal{Q}}(\beta_r + \beta_p, 2\beta_Q K, \beta_{Q,H+1} = 0)$, and similarly $V_{H+1}^k \in \widehat{\mathcal{V}}(\beta_r + \beta_p, 2\beta_Q K, \beta_{Q,H+1} = 0)$.

**Induction step:** Now, suppose the claim holds for $h+1$ and we show it also holds for $h$. We have that

$$
|\hat{Q}_h^k(x,a)| = |\bar{\phi}_h^{k_e}(x,a)^\mathsf{T} w_h^k - \beta_b \|\bar{\phi}_h^{k_e}(x,a)\|_{(\Lambda_h^{k_e})^{-1}}|
$$

$$
\leq |\bar{\phi}_h^{k_e}(x,a)^\mathsf{T}(\theta_h + (\hat{\theta}_h^k - \theta_h) + (\hat{\psi}_h^k - \psi_h)\hat{V}_{h+1}^{k,i} + \psi_h \hat{V}_{h+1}^{k,i})| + \beta_b \|\bar{\phi}_h^{k_e}(x,a)\|_{(\Lambda_h^{k_e})^{-1}}
$$

$$
\leq 1 + \|\hat{V}_{h+1}^{k,i}\|_\infty + \|\bar{\phi}_h^{k_e}(x,a)\|_{(\Lambda_h^{k_e})^{-1}} \left[ \|\hat{\theta}_h^k - \theta_h\|_{\Lambda_h^k} + \|(\hat{\psi}_h^k - \psi_h)\hat{V}_{h+1}^{k,i}\|_{\Lambda_h^k} + \beta_b \right]
$$
$$
\text{(triangle inequality, Cauchy-Schwarz, } \Lambda_h^{k_e} \preceq \Lambda_h^k)
$$

$$
\leq 1 + \beta_{Q,h+1} + (\beta_r + \beta_{p,h} + \beta_b)\|\bar{\phi}_h^{k_e}(x,a)\|_{(\Lambda_h^{k_e})^{-1}}
$$
$$
\text{(induction hypothesis, eqs. (7) and (11))}
$$

$$
\leq 1 + \beta_{Q,h+1} + (\beta_r + \beta_{p,h} + \beta_b)\max_{y\geq 0}[y \cdot \sigma(-\beta_w y + \log K)] \qquad (\bar{\phi}_h^{k_e} \text{ definition})
$$

$$
\leq 1 + \beta_{Q,h+1} + \frac{2\log(eK)}{\beta_w}(\beta_r + \beta_{p,h} + \beta_b) \qquad\qquad\qquad \text{(lemma 18)}
$$

$$
\leq 2 + \beta_{Q,h+1} \qquad\qquad\qquad (\beta_w \geq 2(\beta_r + \beta_{p,h} + \beta_b)\log(eK))
$$

$$
= \beta_{Q,h}.
$$

Additionally, $\beta_b = \beta_r + \beta_p$, $(\Lambda_h^{k_e})^{-1} \preceq I$, $\|\Lambda_h^{k_e}\| \leq 1 + \sum_{k\in[K]}\|\phi_h^k\| \leq 2K$, thus $(\Lambda_h^{k_e})^{-1} \succeq (2K)^{-1}I$, and

$$
\|w_h^k\| = \|\hat{\theta}_h^k + \hat{\psi}_h^k \hat{V}_{h+1}^{k,i}\| \leq K + \beta_Q K \leq 2\beta_Q K = C_w.
$$

We conclude that $\hat{Q}_h^k \in \hat{\mathcal{Q}}(\beta_r + \beta_p, 2\beta_Q K, \beta_{Q,h})$. Since $\pi_h^k \in \Pi(K(\beta_r + \beta_p), 2\beta_Q K^2)$, we also conclude that $\hat{V}_h^k \in \hat{\mathcal{V}}(\beta_r + \beta_p, 2\beta_Q K, \beta_{Q,h})$, proving the induction step and finishing the proof. ∎

**Lemma (restatement of lemma 6).** *Consider the parameter setting of theorem 9. If $\eta_o \leq 1, \beta_w^2 \leq K/(32Hd)$ then $\Pr[E_g] \geq 1 - \delta$.*

**Proof.** Suppose that $\bar{E}_g$ holds. By lemma 11, this occurs with probability at least $1 - \delta$. We show that $\bar{E}_g$ implies $E_g$, thus concluding the proof. Notice that

$$
\pi_h^k(a|x) \propto \exp\left( \eta \sum_{k'=k_e}^{k-1} \hat{Q}_h^{k'}(x,a) \right)
$$

$$
= \exp\left( \sigma(-\beta_w \|\phi(x,a)\|_{(\Lambda_h^{k_e})^{-1}} + \log K) \cdot \left[ \phi(x,a)^\mathsf{T} \sum_{k'=k_e}^{k-1} \eta w_h^k - \eta\beta_b(k - k_e)\|\phi(x,a)\|_{(\Lambda_h^{k_e})^{-1}} \right] \right).
$$

We show by induction on $k \in K_e$ that $\pi_h^k \in \Pi(K(\beta_r + \beta_p), 2\beta_Q K^2)$ for all $h \in [H]$. For the base case, $k = k_e$, $\pi_h^k$ are uniform, corresponding to $w, \beta = 0 \in \Pi(K(\beta_r + \beta_p), 2\beta_Q K^2)$. Now, suppose the claim holds for all $k' < k$. Then by lemma 12 we have that $\hat{Q}_h^{k'} \in \hat{\mathcal{Q}}(\beta_r + \beta_p, 2\beta_Q K, \beta_{Q,h})$ for all $k' < k$ and $h \in [H]$. This implies that $\|\sum_{k'=k_e}^{k-1} \eta w_h^k\| \leq 2\beta_Q K^2$ for all $h \in [H]$, thus $\pi_h^k \in \Pi(K(\beta_r + \beta_p), 2\beta_Q K^2)$ for all $h \in [H]$, concluding the induction step.

Now, since $\pi_h^k \in \Pi(K(\beta_r + \beta_p), 2\beta_Q K^2)$ for all $k \in [K], h \in [H]$, we can apply lemma 12 to get that $\hat{Q}_h^k \in \hat{\mathcal{Q}}(\beta_r + \beta_p, 2\beta_Q K, \beta_{Q,h}), \hat{V}_h^k \in \hat{\mathcal{V}}(\beta_r + \beta_p, 2\beta_Q K, \beta_{Q,h})$ for all $k \in [K], h \in [H]$. Using $\bar{E}_2$ (eq. (11)) we conclude that $E_2$ (eq. (8)) holds, thus concluding the proof. ∎

# B  Technical tools

## B.1  Online Mirror Descent

We begin with a standard regret bound for entropy regularized online mirror descent (hedge). See [Sherman et al., 2023a, Lemma 25].

**Lemma 13.** *Let $y_1, \ldots, y_T \in \mathbb{R}^A$ be any sequence of vectors, and $\eta > 0$ such that $\eta y_t(a) \geq -1$ for all $t \in [T], a \in [A]$. Then if $x_t \in \Delta_A$ is given by $x_1(a) = 1/A \ \forall a$, and for $t \geq 1$:*

$$x_{t+1}(a) = \frac{x_t(a)e^{-\eta y_t(a)}}{\sum_{a' \in [A]} x_t(a')e^{-\eta y_t(a')}},$$

*then,*

$$\max_{x \in \Delta_A} \sum_{t=1}^{T} \sum_{a \in [A]} y_t(a)(x_t(a) - x(a)) \leq \frac{\log A}{\eta} + \eta \sum_{t=1}^{T} \sum_{a \in [A]} x_t(a)y_t(a)^2.$$

## B.2  Value difference lemma

We use the following extended value difference lemma by Shani et al. [2020]. We note that the lemma holds unchanged even for MDP-like structures where the transition kernel $P$ is a sub-stochastic transition kernel, i.e., one with non-negative values that sum to at most one (instead of exactly one).

**Lemma 14 (Extended Value difference Lemma 1 in Shani et al. [2020]).** *Let $\mathcal{M}$ be a (sub) MDP, $\pi, \hat{\pi} \in \Pi_M$ be two policies, $\hat{Q}_h : \mathcal{X} \times \mathcal{A} \to \mathbb{R}, h \in [H]$ be arbitrary function, and $\hat{V}_h : \mathcal{X} \to \mathbb{R}$ be defined as $\hat{V}_h(x) = \sum_{a \in \mathcal{A}} \hat{\pi}_h(a \mid x)\hat{Q}_h(x, a)$. Then*

$$V_1^\pi(x_1) - \hat{V}_1(x_1) = \mathbb{E}_{P,\pi}\left[ \sum_{h \in [H]} \sum_{a \in \mathcal{A}} \hat{Q}_h(x_h, a)(\pi(a \mid x_h) - \hat{\pi}(a \mid x_h)) \right]$$

$$+ \mathbb{E}_{P,\pi}\left[ \sum_{h \in [H]} \ell_h(x_h, a_h) + \sum_{x' \in \mathcal{X}} P(x' \mid x_h, a_h)\hat{V}_{h+1}(x') - \hat{Q}_h(x_h, a_h) \right].$$

*We note that, in the context of linear MDP $\ell_h(x_h, a_h) + \sum_{x' \in \mathcal{X}} P(x' \mid x_h, a_h)\hat{V}_{h+1}(x') = \phi(x_h, a_h)^{\mathsf{T}}(\theta_h + \psi_h \hat{V}_{h+1})$.*

## B.3  Algebraic lemmas

Next, is a well-known bound on harmonic sums [see, e.g., Cohen et al., 2019, Lemma 13]. This is used to show that the optimistic and true losses are close on the realized predictions.

**Lemma 15.** *Let $z_t \in \mathbb{R}^{d'}$ be a sequence such that $\|z_t\|^2 \leq \lambda$, and define $V_t = \lambda I + \sum_{s=1}^{t-1} z_s z_s^{\mathsf{T}}$. Then*

$$\sum_{t=1}^{T} \|z_t\|_{V_t^{-1}} \leq \sqrt{T \sum_{t=1}^{T} \|z_t\|_{V_t^{-1}}^2} \leq \sqrt{2Td' \log(T+1)}.$$

Next, we need the following well-known matrix inequality.

**Lemma 16 (Cohen et al. [2019], Lemma 27).** *If $N \succeq M \succ 0$ then for any vector $v$*

$$\|v\|_N^2 \leq \frac{\det N}{\det M}\|v\|_M^2$$

Next, we need a bound on the Lipschitz constant of the spectral norm of a square-root matrix.

**Lemma 17.** *For any $\lambda > 0$ and matrices $\Lambda, \Lambda' \in \mathbb{R}^{d \times d}$ satisfying $\Lambda, \Lambda' \succeq \lambda I$ we have that*

$$\|\Lambda^{1/2} - \Lambda'^{1/2}\| \leq \frac{1}{2\sqrt{\lambda}}\|\Lambda - \Lambda'\|.$$

**Proof.** Let $\mu$ be an eigenvalue of $\Lambda^{1/2} - \Lambda'^{1/2}$ with eigenvector $x \in \mathbb{R}^d$. Then we have that

$$|x^{\mathsf{T}}(\Lambda - \Lambda')x| = |x^{\mathsf{T}}(\Lambda^{1/2} - \Lambda'^{1/2})\Lambda^{1/2}x + x^{\mathsf{T}}\Lambda'^{1/2}(\Lambda^{1/2} - \Lambda'^{1/2})x|$$
$$= |\mu||x^{\mathsf{T}}(\Lambda^{1/2} + \Lambda'^{1/2})x.$$

Next, notice that $|x^{\mathsf{T}}(\Lambda - \Lambda')x| \leq \|x\|^2\|\Lambda - \Lambda'\|$, and $x^{\mathsf{T}}(\Lambda^{1/2} + \Lambda'^{1/2}) \geq 2\sqrt{\lambda}\|x\|^2$. We thus therefore change sides to get that

$$|\mu| \leq \frac{1}{2\sqrt{\lambda}}\|\Lambda - \Lambda'\|,$$

and since we can take $\mu = \pm\|\Lambda^{1/2} - \Lambda'^{1/2}\|$, the proof is concluded. $\blacksquare$

Finally, we need the following bounds on the logistic function.

**Lemma 18.** *For any $K \geq 1, \beta > 0$ we have that*

$$\max_{y \geq 0}[y \cdot \sigma(-\beta y + \log K)] \leq \frac{2\log(eK)}{\beta}$$

**Proof.** First, if $y' \leq (2\log K)/\beta$ then using $\sigma(y) \in [0, 1]$ we have that

$$y'\sigma(-\beta y' + \log K) \leq y' \leq (2\log K)/\beta,$$

as desired. Now, if $y' \geq (2\log K)/\beta$ then

$$y'\sigma(-\beta y' + \log K) \leq y'\sigma(-\beta y'/2) = \frac{y'}{1 + e^{\beta y'/2}} \leq \frac{y'}{\beta y'/2} = \frac{2}{\beta},$$

where the first inequality also used that $\sigma(y)$ is increasing and the last inequality used that $1 + e^y \geq y$ for all $y \geq 0$. $\blacksquare$

**Lemma 19.** *For any $K \geq 1, z \geq 0$ we have that $\sigma(z - \log K) \leq 2(z^2 + K^{-1})$.*

**Proof.** Recall the logistic function $\sigma(z) = 1/(1 + e^{-x})$ and define the function $g(z) = \sigma(z - \log K) - (z + K^{-1/2})^2$. We show that $g(z) \leq 0$ for all $z \geq 0$. First, notice that

$$g(0) = \sigma(-\log K) - K^{-1} = (K+1)^{-1} - K^{-1} \leq 0.$$

Next, recall that $\sigma'(x) = \sigma(x)\sigma(-x)$ and thus

$$g'(z) = \sigma(z - \log K)\sigma(-z + \log K) - 2(z + K^{-1/2}).$$

Examining $z = 0$ we further have that

$$g'(0) = \sigma(-\log K)\sigma(\log K) - 2K^{-1/2}$$
$$= (K+1)^{-1}(1 + K^{-1})^{-1} - 2K^{-1/2}$$
$$\leq 2[(K+1)^{-1} - K^{-1/2}] \leq 0,$$

where the last two inequalities used $K \geq 1$. Now, we have that

$$g''(z) = \sigma(z - \log K)\sigma(-z + \log K)^2 - \sigma(z - \log K)^2\sigma(-z + \log K) - 2 \leq 0,$$

where the inequality is since $\sigma(z) \in [0, 1]$ for all $z \in \mathbb{R}$. Since $g(0), g'(0) \leq 0$ and $g''(z) \leq 0$ for all $z \geq 0$, we conclude that $g(z) \leq 0$ for all $z \geq 0$. The proof is concluded using the AM-GM inequality. $\blacksquare$

### B.4 Concentration bounds

We give the following Bernstein type tail bound (see e.g., [Rosenberg et al., 2020, Lemma D.4].

**Lemma 20.** *Let $\{X_t\}_{t \geq 1}$ be a sequence of random variables with expectation adapted to a filtration $\mathcal{F}_t$. Suppose that $0 \leq X_t \leq 1$ almost surely. Then with probability at least $1 - \delta$*

$$\sum_{t=1}^{T} \mathbb{E}[X_t \mid \mathcal{F}_{t-1}] \leq 2\sum_{t=1}^{T} X_t + 4\log\frac{2}{\delta}$$

We state the well-known self normalized error bounds for regularized least squares estimation of the rewards and dynamics (see e.g., Abbasi-Yadkori et al. [2011], Jin et al. [2020b]).

**Lemma 21 (reward error bound).** *Let $\widehat{\theta}_h^k$ be as in line 14 of algorithm 1. With probability at least $1 - \delta$, for all $k \geq 1, h \in [H]$*

$$\|\theta_h - \widehat{\theta}_h^k\|_{\Lambda_h^k} \leq 2\sqrt{2d \log(2KH/\delta)}.$$

**Lemma 22 (dynamics error uniform convergence).** *Let $\widehat{\psi}_h^k : \mathbb{R}^{\mathcal{X}} \to \mathbb{R}^d$ be the linear operator defined in eq. (2) inside algorithm 1. For all $h \in [H]$, let $\mathcal{V}_h \subseteq \mathbb{R}^{\mathcal{X}}$ be a set of mappings $V : \mathcal{X} \to \mathbb{R}$ such that $\|V\|_\infty \leq \beta$ and $\beta \geq 1$. With probability at least $1 - \delta$, for all $h \in [H]$, $V \in \mathcal{V}_{h+1}$ and $k \geq 1$*

$$\|(\psi_h - \widehat{\psi}_h^k)V\|_{\Lambda_h^k} \leq 4\beta\sqrt{d \log(K + 1) + 2\log(H\mathcal{N}_\epsilon/\delta)},$$

*where $\epsilon \leq \beta\sqrt{d}/2K$, $\mathcal{N}_\epsilon = \sum_{h \in [H]} \mathcal{N}_{h,\epsilon}$, and $\mathcal{N}_{h,\epsilon}$ is the $\epsilon$−covering number of $\mathcal{V}_h$ with respect to the supremum distance.*

## B.5  Covering numbers

The following results are (mostly) standard bounds on the covering number of several function classes.

**Lemma 23.** *For any $\epsilon > 0$, the $\epsilon$-covering of the Euclidean ball in $\mathbb{R}^d$ with radius $R \geq 0$ is upper bounded by $(1 + 2R/\epsilon)^d$.*

**Lemma 24.** *Let $\mathcal{V} = \{V(\cdot; \theta) : \|\theta\| \leq W\}$ denote a class of functions $V : \mathcal{X} \to \mathbb{R}$. Suppose that any $V \in \mathcal{V}$ is L-Lipschitz with respect to $\theta$ and supremum distance, i.e.,*

$$\|V(\cdot; \theta_1) - V(\cdot; \theta_2)\|_\infty \leq L\|\theta_1 - \theta_2\|, \quad \|\theta_1\|, \|\theta_2\| \leq W.$$

*Let $\mathcal{N}_\epsilon$ be the $\epsilon$−covering number of $\mathcal{V}$ with respect to the supremum distance. Then*

$$\log \mathcal{N}_\epsilon \leq d \log(1 + 2WL/\epsilon)$$

**Proof.** Let $\Theta_{\epsilon/L}$ be an $(\epsilon/L)$-covering of the Euclidean ball in $\mathbb{R}^d$ with radius $W$. Define $\mathcal{V}_\epsilon = \{V(\cdot; \theta) : \theta \in \Theta_{\epsilon/L}\}$. By lemma 23 we have that $\log|\mathcal{V}_\epsilon| \leq d \log(1 + 2WL/\epsilon)$. We show that $\mathcal{V}_\epsilon$ is an $\epsilon$-cover of $\mathcal{V}_\epsilon$, thus concluding the proof. Let $V \in \mathcal{V}$ and $\theta$ be its associated parameter. Let $\theta' \in \Theta_{\epsilon/L}$ be the point in the cover nearest to $\theta$ and $V' \in \mathcal{V}$ its associated function. Then we have that

$$\|V(\cdot) - V'(\cdot)\|_\infty = \|V(\cdot; \theta) - V(\cdot; \theta')\|_\infty \leq L\|\theta - \theta'\| \leq L(\epsilon/L) = \epsilon. \qquad \blacksquare$$

