# OpenReview forum: "Warm-up Free Policy Optimization: Improved Regret in Linear Markov Decision Processes"
_NeurIPS.cc/2024/Conference — NeurIPS 2024 poster_

### Official Review · Reviewer_gZJ8 · 2024-07-04

**Soundness:** 2
**Presentation:** 4
**Contribution:** 3
**Rating:** 6
**Confidence:** 3

**Summary:**

This paper improves the Policy Optimization methods for learning MDP by eliminating the undesired warm-up phase and replacing it with a simple and efficient contraction mechanism. For linear MDP, it is shown that the proposed Policy Optimization algorithm achieves regret with improved dependence on problem parameters (the horizon and function approximation dimension) under the settings of adversarial losses with full-information feedback and stochastic losses with bandit feedback. The contraction mechanism serves the purpose of ensuring the Q-value estimates are bounded and yield simple policies in the sense of efficient computation and easy implementation. The regret bound improves upon the best known ones for policy optimization.

**Strengths:**

**Significance**:

**1.** This paper improved the regret bound for policy optimization algorithms for linear MDPs. Specifically, the CFPO algorithm achieves a $\sqrt{H^3 d}$ improvement over the best known regret for adversarial setting, and matches the performance of the value-iteration based algorithm.

**2** The policy optimization algorithm is computationally more efficient and easier to implement. This is because the algorithm uses all samples and is reward-aware (thus no waste of information).

**Clarity**: This paper is clearly written.

**1.** All necessary related work is properly discussed in my opinion.

**2.** The algorithm is introduced with a clear explanation. In section 4, a detailed walk-through of the CFPO algorithm is given with explicit items comparing the new algorithm with existing one, making it clear why the new one enjoys improved performance. A simple example is given (bottom of page 6) to demonstrate why reward-awareness is beneficial in PO.

**Weaknesses:**

There is no major technical flaw detected in this paper.

Weakness:

There is no experimental result to corroborate the theoretical results. Though PO algorithms are a bit more complicated to implement than the value-iteration algorithms in my opinion, for linear MDP, a numerical simulation or simple synthetic experiment should not seem too difficult.

And I think experimental results is especially important for this work, since it claims the deviced algorithm is easier to implement compared to existing PO algorithms. Therefore, experimental results verifying the correctness of the improved regret and the easier implementation than existing PO baselines are anticipated. For example, for the improved regret, it would be good to see from experiment that the dependence on the problem parameter is indeed improved. A very simple synthetic environment should be enough.

**Questions:**

Please see Weaknesses.

**Limitations:**

Yes.

---

> ### Author Rebuttal · Authors · 2024-08-02
>
> Thank you for the time and effort put into the review.
>
> Regarding synthetic experiments, we are not aware of existing benchmarks that are specific to linear MDPs rather than tabular ones. However, in tabular MDPs the contraction is not necessary and thus we do not expect to see improvement. We agree that coming up with interesting experimental setups for linear MDPs would be valuable to the community but leave this to future research. Please note that most papers on regret minimization in MDPs do not include experiments.

---

> > ### Comment · Reviewer_gZJ8 · 2024-08-12
> >
> > I would like to thank the authors for the rebuttal. I have no additional concerns and thus raised the score.

---

### Official Review · Reviewer_Gemc · 2024-07-04

**Soundness:** 3
**Presentation:** 3
**Contribution:** 3
**Rating:** 7
**Confidence:** 4

**Summary:**

This paper equips the rare-switching mechanism with a novel feature shrinkage technique to achieve efficient policy optimization (PO) for linear MDPs with adversarial losses or bandit feedback. By shrinking features in directions of high uncertainty, the authors show that the proposed algorithm has its regret optimal in terms of $K$, the number of episodes, up to a logarithmic factor. Compared with prior work on PO for adversarial linear MDPs, the proposed algorithm does not invoke sophisticated black boxes or incur any $\text{poly}(d, H)$ burn-in cost and enjoys a regret upper bound with lower dependence on the horizon $H$ and the ambient dimension $d$.

**Strengths:**

1. The proposed feature shrinkage technique, which enables the estimated state-action value to be bounded without resort to truncation techniques, is simple and does not incur any additional statistical or computational overhead up to constant factors.
2. The authors notice that the extended value difference lemma [1] is applicable (for the proposed contracted subMDP) even if the transition kernel is sub-stochastic, which is a key observation that might be of independent interest and useful for the reinforcement learning (RL) community.


[1] Shani, L., Efroni, Y., Rosenberg, A., & Mannor, S. (2020, November). Optimistic policy optimization with bandit feedback. In International Conference on Machine Learning (pp. 8604-8613). PMLR.

**Weaknesses:**

## Weaknesses

1. The authors should make it clear in the main text whether the analysis or the final regret bound depends on $|\mathcal{X}| < \infty$ in any significant way. As far as I can tell, the regret bound does not rely on the finite $\mathcal{X}$ assumption, but the authors should further clarify this point.
2. Line 266: [1] does not utilize any feature shrinkage technique, so it is confusing and not adequate to mention "the corresponding claim" and use the $\bar{\phi}^{k_e}$ notation in equation (6) without further explanation.
3. Line 49: Though the results in this paper is significant, it is way too assertive to say that no simpler algorithm with lower regret exists for adversarial linear MDPs. The authors should at least elucidate their conjecture about the optimality of the proposed algorithm in a more appropriate manner.
4. Minor issue: Technically speaking, the two $\sum_{h}$ signs between Line 276 and 277 should not be put outside the expectation in that your expectation is taken with respect to the randomness of a trajectory.
5. Between Line 151 and 152: the $\Delta P(x_h, a_h)$ part seems to be a typo
6. The key connection between the proposed technique, especially Lemma 18, and the boundedness of the estimated state-action value function (i.e., the arguments between Line 549 and 550) is not explained in any way in the main text. (BTW, in the proof of Lemma 18, it seems that $K\geq 1$ should be changed to $K\geq \mathrm{e}$)
7. Minor typo on Line 265: $\succ$ -> $\leq$
8. Minor typo on Line 579: $\hat{V}$ -> $\hat{V}_h$

[1] Sherman, U., Cohen, A., Koren, T., & Mansour, Y. (2023). Rate-optimal policy optimization for linear markov decision processes. arXiv preprint arXiv:2308.14642.

**Questions:**

1. Any reference on the proposition mentioned on Line 134-135 that the Markov $\pi^*$ in hindsight is optimal among all policies?

---

> ### Author Rebuttal · Authors · 2024-08-02
>
> Thank you for the thorough review and helpful comments, we will incorporate them in our revision. The following responds to your individual points:
> 1. Finite $\mathcal{X}$: You are correct, the regret does not depend on the assumption that $\mathcal{X}$ is finite. In short, the assumption is purely technical as it helps avoid measure theoretic notations and allows for a cleaner and more approachable analysis. The explanation for this was deferred to an existing paper but we agree that it should be included in our paper for the sake of completeness.
> 2. L266: Thank you for pointing this out, we will revise this explanation in the final version. Our intention was to say that the reward free warm-up in [1] gives a guarantee of the form in eq.(6) but where $\bar{\phi}_h^{k_e}(x,a)$ is replaced with $\phi(x,a) \mathbb{1}(x \in \mathcal{Z}_h)$. The overall implication is that, through the lens of contracted sub-MDPs, the reward free warm-up would give an overly conservative contraction that incurs additional cost compared to our approach.
> 3. L49: The purpose of this phrase was to say that while our rates are not minimax, improving them further likely requires more delicate algorithmic techniques that are not well-understood for PO even in tabular MDPs. In particular, we are not aware of any method besides the mentioned variance reduction technique that achieves better rates. We will soften the phrasing of the final version such that it is more clear that this is a conjecture rather than a proven fact.
> 4. L276-277: The expectation is indeed over trajectories. However, because their length is fixed, we can take the $\sum_h$ outside due to the linearity of the expectation. As you pointed out, the arguments work either way.
> 5. L151-152 $\Delta P$: We overloaded notation here (perhaps excessively). We’ll disambiguate it in the final version.
> 6. Key connection… boundedness of $\hat{Q}$: We omitted this due to space constraints. We’ll include an explanation in the final version. Thanks for pointing out that $K \ge e$ in Lemma 18.
> 7. Minor typos (points 7-8): Thanks for finding these! We’ll fix them for the final version.
> 8. Markov $\pi^\star$ is optimal even among history dependent policies: There are probably several sources for this claim. For example, Reinforcement Learning:  Foundations (p.56)  by Shie Mannor, Yishay Mansour, and Aviv Tamar.

---

> > ### Comment · Reviewer_Gemc · 2024-08-12
> > **Thanks.**
> >
> > The authors have appropriately answered all questions I asked. I will keep my positive evaluation and recommend this paper for acceptance.

---

### Official Review · Reviewer_x3ta · 2024-07-12

**Soundness:** 3
**Presentation:** 3
**Contribution:** 3
**Rating:** 6
**Confidence:** 3

**Summary:**

This paper presents a new policy optimization algorithm called Contracted Features Policy Optimization (CFPO) for reinforcement learning in linear Markov Decision Processes (MDPs). The key contribution is eliminating the need for a costly warm-up phase used in previous state-of-the-art methods, while achieving improved regret bounds.

**Strengths:**

The paper addresses a significant issue in reinforcement learning theory, improving on recent findings for policy optimization in linear MDPs.

The proposed CFPO algorithm eliminates the need for a separate warm-up phase, offering a substantial practical advantage over previous methods.

The regret bounds show improvement compared to prior work, achieving $O(\sqrt{H^4 d^3 K})$ regret, which represents a $\sqrt{H^3 d}$ enhancement.

The analysis introduces novel techniques, such as the contracted MDP concept, which may have broader applications in the field.

**Weaknesses:**

While the regret bound in this paper shows a better dependency on K compared to previous works, it has a worse dependency on d and H. Although K is the most critical factor, it might be more appropriate to discuss the improvement in regret when K exceeds a certain threshold.

The paper is purely theoretical, lacking experimental results to validate the practical performance of the proposed algorithm. More discussion on practical implications and potential applications would be beneficial.


Despite the improvement, there remains a gap between the upper and lower bounds for this problem. Further discussion on potential approaches to address this gap would be valuable. For example, can previous variance reduction techniques be applied to this method?

 The paper could be strengthened by adding a conclusion section to summarize the key findings

**Questions:**

na

**Limitations:**

The authors should include a more detailed discussion of the study's limitations in the conclusion part.

---

> ### Author Rebuttal · Authors · 2024-08-02
>
> Thank you for the time and effort put into the review. The following addresses the points made in your review:
> 1. Regarding dependence on $K,d,H$: Are you referring to the additive regret term that is logarithmic in $K$? As you mentioned, most works assume that $K$ is the dominant factor and thus this term is omitted. For our algorithm it becomes a low order term when $K \ge \sqrt{H^4 d^3}$. Notice that the regret bound of $\sqrt{H^4 d^3 K}$ is trivial, i.e., $ \ge KH$, when $K \le \sqrt{H^2 d^3}$ and thus the additional term is non-trivial only between these two values. The regret bound of Sherman et al. (2023) is non-trivial only for $K \ge \sqrt{H^5 d^4}$ and thus our regret bound is equivalent or better for all values of $K,d,H$. Overall, you bring up a subtle point. If you think this will improve the paper then we are willing to explain this in the final version.
> 2. Discussion: We will add a discussion about potential applications, limitations, and the gap from the lower bound. We conjecture that it is possible to use variance reduction methods for PO. However, this seems quite complicated and thus left for future research.

---

> > ### Comment · Reviewer_x3ta · 2024-08-13
> >
> > Thanks for your rebuttal. I hope the authors will consider incorporating these clarifications in their revision. I will maintain my positive assessment.

---

### Official Review · Reviewer_JkeU · 2024-07-13

**Soundness:** 3
**Presentation:** 3
**Contribution:** 3
**Rating:** 7
**Confidence:** 4

**Summary:**

This paper studies online learning for linear MDPs with stochastic and adversarial full-information losses. The authors propose a new contraction mechanism, avoiding the costly initial exploration phase in previous papers and achieving a better regret bound.

**Strengths:**

1. This paper studies an important problem with improved regret bounds.
2. The new algorithm gets rid of a costly initial exploration phase in previous papers using new techniques, which may give more insights for future works in both theory and practice.

**Weaknesses:**

I do not see obvious weakness. For writing suggestions, I feel it is beneficial to discuss more about the difficulty of bounding covering numbers when applying PO for linear MDP. Although [Zhong and Zhang 2023] and [Sherman et al. 2023a] gave comprehensive discussions, it is good to discuss more in this paper to ensure the readers have a whole picture (e.g. discuss more why clipping Q-functions does not lead to good covering numbers for policy class).

**Questions:**

Currently, I do not have questions.

**Limitations:**

Yes.

---

> ### Author Rebuttal · Authors · 2024-08-02
>
> Thank you very much for the positive comments. We agree that reiterating the explanations on the effect of clipping on the covering number will make the paper more self-contained and we will include it in the final version of the paper.

---

### Decision · Program_Chairs · 2024-09-25

**Decision:**

Accept (poster)

**Comment:**

The paper introduces the CFPO algorithm for linear MDPs, improving upon prior work by removing the warm-up phase and achieving better regret bounds. Reviewers praised its theoretical contributions, particularly the novel contraction mechanism, and its potential impact on reinforcement learning. However, they raised concerns about the lack of experimental validation and suggested clarifying some technical dependencies on problem parameters.

In rebuttal phase, the authors clarified the points raised, addressed minor presentation issues, and acknowledged the need for future experiments. Due to the paper’s strong theoretical results, reviewers were overall positive after the rebuttal. I recommend acceptance of this paper as a poster presentation.